



# Impact Ionization Double Peaks Analyzed in High Temporal Resolution on Solar Orbiter

Samuel Kočiščák[1], Ingrid Mann[1], Nicole Meyer-Vernet[2], Audun Theodorsen[1], Jakub Vaverka[3], and Arnaud Zaslavsky[2]

[1]Department of Physics and Technology, UiT The Arctic University of Norway, 9037 Tromsø, Norway
[2]LESIA, Observatoire de Paris, Université PSL, CNRS, Sorbonne Université, Université de Paris, Paris, France
[3]Faculty of Mathematics and Physics, Charles University, Prague, Czech Republic

**Correspondence:** Samuel Kočiščák (samuel.kociscak@uit.no)

**Abstract.**

Solar Orbiter is equipped with electrical antennas performing fast measurements of the surrounding electric field. The antennas register high velocity dust impacts through the electrical signatures of impact ionization. Although the basic principle of the detection has been known for decades, the understanding of the underlying process is not complete, due to unique mechanical

and electrical design of each spacecraft and the variability of the process.

We present a study of electrical signatures of dust impacts on Solar Orbiter's body, as measured with Radio and Plasma Waves electrical suite. A large proportion of the signatures present a double-peak electrical waveforms in addition to the fast pre-spike due to electron motion, which are systematically observed for the first time. We believe this is due to Solar Orbiter's unique antenna design and a high temporal resolution of the measurements. The double peaks are explained as due to two

distinct processes. Qualitative and quantitative features of both peaks are described. The process for producing the primary peak is known for a long time and the process for producing the secondary peak was proposed before (Pantellini et al., 2012a) for Solar Terrestrial Relations Observatory (STEREO), although the corresponding delay of $100\,\mu s - 300\,\mu s$ against the primary peak was not observed until now.

The primary peak's amplitude is believed to be the better measure of the impact-produced charge and suggests that the

typical amount is around $8\,pC$. The observed asymmetry between the primary peaks measured with individual antennas is quantitatively explained as electrostatic induction. A relationship between the primary and the secondary peaks' amplitudes is found to be non-linear and the relation is partially explained with a model for electrical interaction through the antennas' photoelectron sheath.

## 1 Introduction

Since their first in situ observation, interplanetary dust particles were observed with not only specialized instruments, but also as byproducts of other measurements, making dust detections much more abundant. One promising and actively discussed option for auxiliary dust detection of recent years is impact ionization detection with electrical antennas (Meyer-Vernet (2001); Mann et al. (2014) and references therein). When spacecraft collides with a dust grain at a relative velocity exceeding a few



km/s, the impact releases free charge due to the high energy density present on the impact site (Friichtenicht, 1964). The
released charge is quasi-neutral, yet the present fields often act to separate positive and negative constituents quickly, allowing
for its effective detection through the signature in the electric field measurement, once separated. How exactly the detection is
done depends greatly on the spacecraft's properties, surrounding environment, impact site and detecting apparatus. In any case,
the preturbation of the electric field stays present for less than $1\,\mathrm{ms}$, while the process of charge separation takes even much
shorter. Therefore, fast electrical measurements are needed in order to observe the process closely.

Solar Orbiter is one of the first (Bale et al., 2016; Maksimovic et al., 2020; Mann et al., 2019) missions to include a wave
analyzer suite designed with dust detection in mind. Dust impact events are readily recognized based on a characteristic peak
(Zaslavsky et al., 2021; Kvammen et al., 2023), yet the analysis and the interpretation of the recorded signals is made difficult
by unclear dependence of the process on spacecraft properties, which is also an issue with other spacecraft conducting similar
detection (Zaslavsky et al., 2012; Malaspina et al., 2014; Vaverka et al., 2017; Ye et al., 2019; Page et al., 2020; Ye et al., 2020;
Zaslavsky et al., 2021; Kellogg et al., 2021; Racković Babić et al., 2022). In the present paper we report the first observation
of a double-peak structure (in addition to the fast electron pre-spike) associated with dust impacts recorded with electrical
antennas. The double peak structure is explained as caused by two charge collection processes happening simultaneously or in
a quick succession, and analyzed as such.

We structure the article as follows: In this section, we present Solar Orbiter as a dust detector. We inspect the data and
describe our findings in Section 2. In Section 3, we describe the electrical process theoretically and with quantitative estimates
as due to two processes. In Sections 4 and 5, we focus on primary and secondary peaks respectively. We show that the primary
peaks are understood with current knowledge and we discuss potential explanations for the secondary peaks. We summarize in
Section 6.

## 1.1 Solar Orbiter as a dust detector

Solar Orbiter is a three-axis stabilized spacecraft, launched on February 10[th] 2020, orbiting the Sun, with an aphelion near
$1\,\mathrm{AU}$ and a perihelion shrinking from $0.5\,\mathrm{AU}$ to currently $0.28\,\mathrm{AU}$. Solar Orbiter remained close to the ecliptic plane so far,
but will be gaining orbital inclination gradually, starting in 2023 and reaching the maximum inclination of $24^\circ$ and possibly
$33^\circ$ in late 2020s.

The area of the Solar Orbiter's body and shield combined is $\approx 28.4\,\mathrm{m}^2$, where $\approx 7.4\,\mathrm{m}^2$ of this value is taken by the heat
shield front side, the rest is the spacecraft's body and the heat shield backside (ESA, 2023). In addition, the backside of the solar
panels is conductive and coupled to the body, which adds another $15.1\,\mathrm{m}^2$ (Zaslavsky et al., 2021). The spacecraft therefore
provides $\approx 43.5\,\mathrm{m}^2$ of surface sensitive to dust impacts, with anisotropic distribution. We note that the areas are based on a
simplified description of the spacecraft as a cuboid with a heat shield, while a portion of the area is covered by insensitive
surfaces. Other sensitive surfaces may contribute to the area besides the cuboid. The heat shield is made of calcium phosphate
coated titanium, while the body is covered with various metallic and non-metallic materials. Which materials are exposed
definitely plays a role for the distribution of impact amplitudes and this is worthy of future investigation.



### 1.1.1 Radio and Plasma Waves instrument

Radio and Plasma Waves instrument (RPW) is a combined electric and magnetic suite for an in situ study of fields and waves (Maksimovic et al., 2020). It provides fast electrical measurements with its three rigid conical nickel cobalt alloy antennas, which enable detection of dust impact events. Each of the antennas is $6.5\,\mathrm{m}$ long with the near-base diameter of $38\,\mathrm{mm}$ and they lie in one plane recessed approximately $1\,\mathrm{m}$ behind the heat shield. Though dust impact events might be identifiable in the +electrical spectra, the Time Domain Sampler subsystem (TDS) of RPW is the key to a robust analysis (Zaslavsky et al., 2021), since the dust impacts are solitary pulse events that provide little information in spectra.

### 1.1.2 Radio and Plasma Waves data

The three RPW electrical antennas measure in various configurations: monopole, dipole, and mixed. In the monopole configuration, abbreviated SE1, antennas measure voltage against the spacecraft body — this configuration is in principle best suited for dust detection, as the dust impacts' influence on the body potential is of interest. In the dipole mode (DIFF1), antennas measure against each other, which has the benefit of the largest effective length for the electrical fields study, but the measurement is nearly insensitive to the changes of the potential of the body. Nonetheless, dust impacts were identified in dipole measurements before and can be identified in DIFF1 measurements of Solar Orbiter, given that the impact influences the potential of an antenna. DIFF1 measurement also provides a redundant information on electric fields, as the three antennas lie in a plane, hence only two components could be measured. In the mixed mode (XLD1), the three channels are occupied by two dipoles and a monopole, which in principle retains benefits of both aforementioned configurations, as both monopole and dipole signals could be reconstructed. For more detailed description, see Appendix A. The XLD1 mode is the one the instrument spends the most time in.

The RPW records electrical waveforms with $6.25\,\%$ duty cycle, that is the first $62.5\,\mathrm{ms}$ of every second. In trigger mode, the on-board algorithm decides whether to keep each of the recordings, based on the maximum amplitude observed within the window. Several hundreds out of approx. 86400 windows a day are stored and transmitted. The on-board algorithm also classifies the stored waveforms into different phenomena categories, one of which is the dust impact. The on-board algorithm, however, does not achieve the precision and accuracy of ground based classifications. In a recent paper, Kvammen et al. (2023) re-did the classification with machine-learning techniques. In present paper, this data is used (Kvammen, 2022).

For the purpose of waveform analysis and plotting in the present work, the raw data is altered by a sequence of digital filters. As a result, the waveforms are trusted in the bandwidth of $500\,\mathrm{Hz} < f < 70\,\mathrm{kHz}$. For a comprehensive description, consult Appendix B.

## 2 Observation of impact ionization on Solar Orbiter

Solar Orbiter's RPW electrical antennas (Maksimovic et al., 2020) are similar in terms of construction and the sampling rate to the electrical suite of Solar Terrestrial Observatory (STEREO) /Waves (Bale et al., 2008). The antennas are rigid thick poles,





with the difference that in the case of STEREO/Waves, the bases of the three orthogonal antennas are physically close to one another, while in the case of Solar Orbiter/RPW, the three antennas lie in one plane and their bases are physically distant, with
the spacecraft's body between them. Nevertheless, the systems' semblance suggests comparable capabilities for dust detection. Therefore, in this section we will present and examine the dust data acquired with Solar Orbiter/RPW, building on the results of and comparing to STEREO/Waves.

## 2.1  Single and triple hits

STEREO had observed two kinds of dust impact events, so called single-hits and triple-hits. The difference is that the triple hits
are observed similarly strong on all three channels, which suggests that most of the process takes place on the common ground the channels measure against, rather than on each of the antennas (Zaslavsky et al., 2012). The single hits were reportedly produced by nanodust impacts, which were observed on both STEREO and Cassini with similar fluxes (Schippers et al., 2014, 2015; Meyer-Vernet et al., 2017) when the solar wind electric field focused them towards the ecliptic (Juhász and Horányi, 2013) — a condition that stopped after 2012 (Le Chat et al., 2015). Since they produce small voltages, they were only
observed on the antenna lying within the impact cloud, whose voltages could be amplified (Pantellini et al., 2012a; Zaslavsky et al., 2012), and their flux was several orders of magnitude larger than that of beta particles and much more variable, as predicted by Mann et al. (2007). Although STEREO-like single hits are not expected to return until after 2024 (Poppe and Lee, 2020, 2022), it is useful to compare the channels' amplitudes to one another and we will keep using the terms single and triple hits for Solar Orbiter events, where appropriate. We compare the amplitudes using the ternary plot of channels' maxima,
that is the highest amplitude of the voltage between the antenna and the body. The ternary plot is the plot in an equilateral triangle, in which the position in the triangle corresponds to the relative contribution of the three contributors to the sum. In our case, ternary plots are normalized to the sum of three channel maxima for an impact, showing a relative amplitude of the three channel maxima, see Fig. 1.





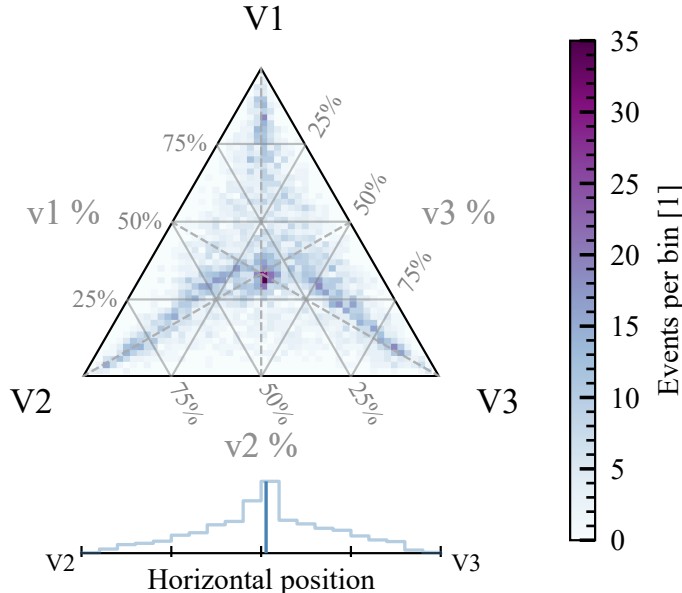

**Figure 1.** Heat-map in the ternary plot for the channel maxima ($VX = ant_X - body$) for all the events identified as dust impacts. 4534 data points contribute to the heat-map.

We see that many events lie near the center, which corresponds to a similar response on all three antennas. However, many

events lie towards the corners as well, especially near the triangle's medians, which implies an amplitude in one channel higher than in the other two channels, which are in turn nearly equal to one another. This suggests that a process concerning antenna might be present — similar to the conclusion made for STEREO's single hits (Zaslavsky et al., 2012; Pantellini et al., 2012a). The spacecraft has a rough lateral mirror symmetry between antennas 2 and 3, while antenna 1 lies in the plane of symmetry. We see a small preference of antenna 3 against antenna 2, which is to be expected, since the antenna 3 is closest to the ram

direction, while antenna 2 is close to the anti-ram. Hence, we expect more impacts in the vicinity of antenna 3, compared to the antenna 2. The schematic view of the three antennas with respect to the spacecraft body is shown in Fig. F1. We also see that double hits (strong in two and weak in one channel) are not very frequent, but clearly the pair of antenna 3 and antenna 1 is the most prevalent for such hits. This is also to be expected given the direction of the ram. Note that this is a crude representation as it only accounts for the global positive maxima, which is an imperfect measure of impact location. Overall, the preference

for ram direction is apparent and a process concerning antennas is hinted through the presence of single hits.

## 2.2 Waveforms inspection

Upon inspection of the processed signals (see Appendix B) recorded in monopole (SE1) mode (see Fig. 2), we see that many of the waveforms show the following structure: a simultaneous peaks of similar amplitudes in all three of the channels (Fig. 2 (a), let us denote the peak a *primary peak*), often followed by a *secondary* peak of a different amplitude and delay in each




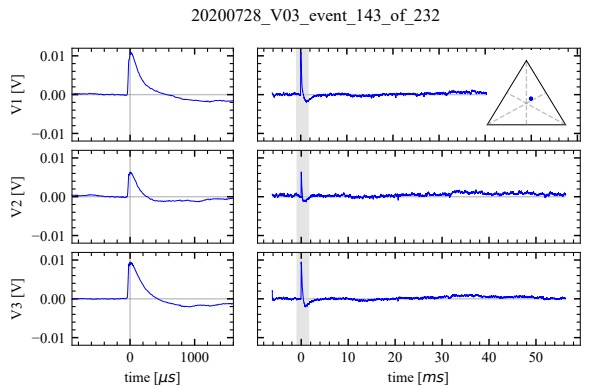

(a) A clear triple-hit: simultaneous and with similar amplitude in all three channels.

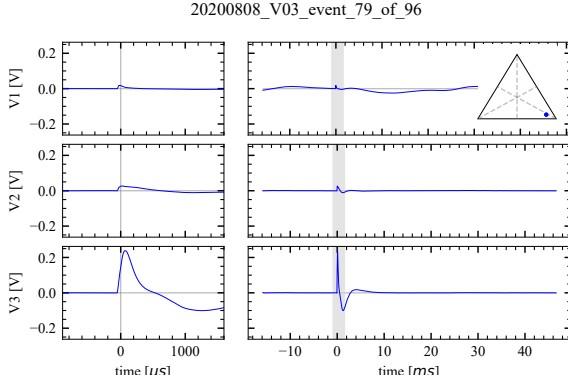

(b) Channel V3 shows larger amplitude, compared to channels V1 and V2. A relative delay of $\approx 50\,\mu s$ is present.

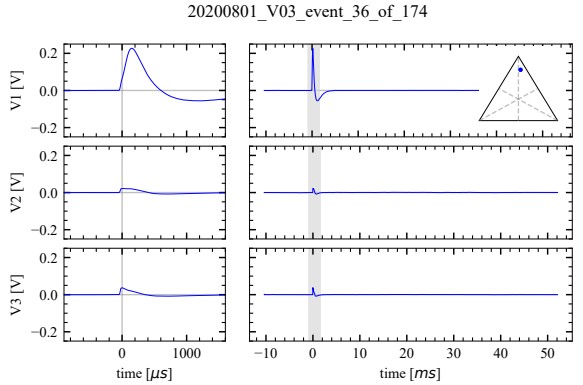

(c) Channel V1 shows larger amplitude, compared to channels V2 and V3. A relative delay of $\approx 150\,\mu s$ is present.

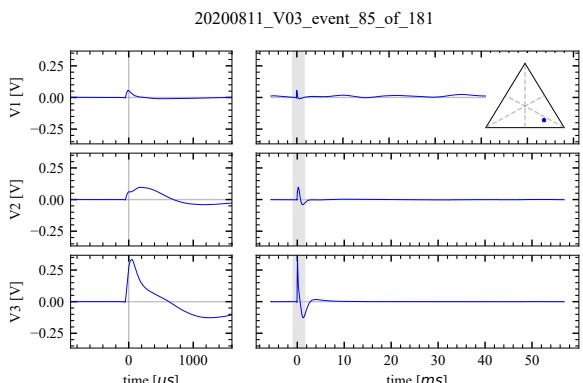

(d) A common primary peak is visible in channels V1 and V2, a secondary peak is present in V2, a larger amplitude and a delay are present in V3.

**Figure 2.** Dust impact events, recorded in true monopole (SE1) mode, processed (see Appendix B). The voltages are shown as $VX = ant_X - body$. The triangular insets show the corresponding location of the event on the amplitude ternary plot, consult Fig. 1. The left-hand-side shows detail of the shaded portion of the right-hand-side, which in turn shows the whole recording of 62ms.



channel, not always present in all of the channels (Fig. 2 (b,c,d)). Sometimes one of the channels shows a more prominent peak instead of one of the primary peak (Fig. 2 (d)). It seems reasonable to explain these cases as the secondary peak following shortly after the primary peak, hence overshadowing the primary peak. Since it is often that just one of the channels shows a secondary peak much stronger than the primary peak (Fig. 2 (b,c,d)), we identify the often-seen single hits as being due to the secondary peak (see Fig. 1 and the corresponding discussion). The two-peak structure is clearly present in many of the impacts

($\approx 50\%$) and even more are consistent with the pattern. To the best of our knowledge, this is the first time when such clear double-peak structures in the impact signals were observed. For separate ternary plots for the impacts that do and that do not show double-peak structure, see Appendix C.

     Signals recorded in mixed (XLD1) mode, decomposed to the monopole channels (see Appendix A) and processed the same way as monopole signals (see Appendix B) fit the description outlined in the previous paragraph as well (see Fig. 3). This is

not surprising, given that the information retained in XLD1 data is virtually the same, except for saturation levels and, to a minor extent, bandwidth. This however confirms that we are justified to treat decomposed XLD1 data the same way as one would treat the monopole signals.

     In addition to the primary and the secondary peak, there is often a negative pre-spike present in the waveforms, immediately predeceding the main signal. We believe this to be due to electron dynamics and we will address it later on.

There is a post-impact negative overshoot present in many of the recordings shown in plots in Figs. 2 and 3. One possible explanation for this behavior was developed and described in Zaslavsky (2015) as due to a partial collection of the electrons by antennas, that have longer discharge time constant $\tau_{RC}$ compared to the spacecraft's body. More generally, the behavior is the same even if the antenna is charged by a different process than one described by Zaslavsky (2015), that is, the charge does not have to come directly from the impact plasma. We will not pursue the explanation now, as the tails of the impacts are

generally on the edge or outside of the trusted bandwidth, that is $f < 500\,\mathrm{Hz}$, of $\tau \approx 2\,\mathrm{ms}$. Let us only note that even though the overshoots are likely distorted and out of scope of this paper, they are likely at least partially physical, as similar overshoots were observed on STEREO (Zaslavsky, 2015) and Parker Solar Probe (Kellogg et al., 2021).

## 2.3    Features extraction

For the present analysis, we used the CNN-refined data described in Kvammen et al. (2023), decomposed into monopole

signals. In order to describe the events of interest only, that is the body impacts onto sunlit metallic parts conductively coupled to the spacecraft's body, we employ the following filtering criteria: only the impacts of a maximum amplitude below $0.3\,\mathrm{V}$ that are predominantly positive in all the monopole channels were analyzed. The upper limit of $0.3\,\mathrm{V}$ is employed in order to avoid reaching the saturation level. We note that predominantly negative pulses produced by antenna hits are also present in the data, yet out of scope of present work, as the electrical process is different for these. Besides, we disregarded the signals

captured very near the beginning or the end of the recording window. After applying these criteria, we are left with $\gtrsim 50\%$ of the waveforms in the CNN dataset.

     We are interested in the following parameters: amplitude of the primary peak, electron prespike presence and amplitude, and secondary peaks' presence and amplitudes, and the primary peak's rise and decay times; where the former two peaks (electron



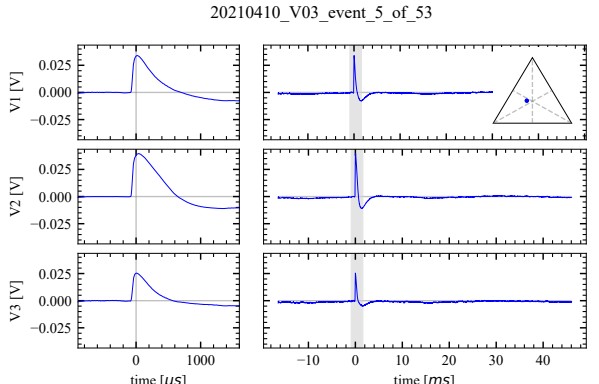

(a) A clear triple-hit: simultaneous and with similar amplitude in all three channels.

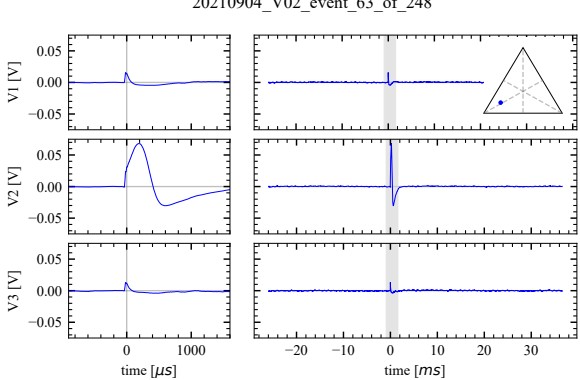

(b) Channel V2 shows larger amplitude, compared to channels V1 and V3. A relative delay of $\approx 200\,\mu s$ is present.

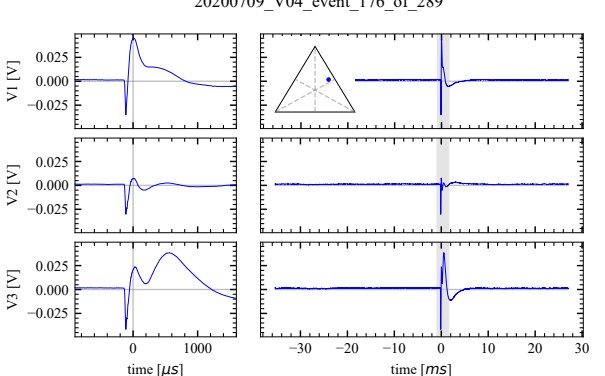

(c) A common primary peak is visible all the channels, a secondary peak is present in V3, with hints of it in V1 and V2. A negative prespike is clearly present.

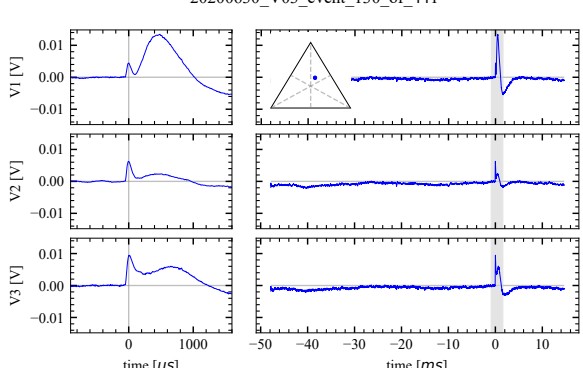

(d) A common primary peak is visible in all three channels, a larger amplitude and delay peak is present in V1. Hints of secondary peaks are present in V2 and V3 with different delays.

**Figure 3.** Dust impact events, monopoles reconstructed from signals recorded in hybrid monopole / dipole (XLD1) mode, processed (see Appendix B). The voltages are shown as $VX = ant_X - body$. The triangular insets show the corresponding location of the event on the amplitude ternary plot, consult Fig. 1. The left-hand-side shows detail of the shaded portion of the right-hand-side, which in turn shows the whole recording of 62ms.





and primary peaks) are assumed to be common in all three channels, the latter (secondary) is analyzed channel-wise. For a
comprehensive description how these are extracted, the reader is referred to the Appendix D.

## 3   Dust impact pulse and process description

Given the previous literature (Friichtenicht (1964); Auer and Sitte (1968); Gurnett et al. (1983); Zaslavsky et al. (2012);
Pantellini et al. (2012a); Meyer-Vernet et al. (2014, 2017); Vaverka et al. (2017); Mann et al. (2019); Ye et al. (2019); Kočiščák
et al. (2020); Kellogg et al. (2021); Shen et al. (2021); Racković Babić et al. (2022); Shen et al. (2023)) and what we observe
in case of Solar Orbiter's RPW data, we formulate a following simplified outlook on the process.

Since the spacecraft is virtually always in the sunlight, photoelectrons are released from its body, leading to a positive
charge of the most of the spacecraft's body. Upon a hypervelocity dust impact onto the spacecraft body, quasi-neutral charge
is released. In case of a spacecraft's body hit, measurement of the spacecraft's antennas potential against its body ($\Phi_{ant} -$
$\Phi_{body}$) show an evolution of the voltage difference, summarized on different timescales as follows. The phases 1. – 5. are also
visualized in Fig. 4.

1. The impact. A quasi-neutral cloud is born in the near vicinity of the spacecraft. Neglecting a usually small charge
   possibly carried by the incident dust grain, no change is induced to the spacecraft's potential due to the impact, as the
   newborn cloud is quasi-neutral and all the charged particles remain in near vicinity of each other and therefore have no
   net influence on the potential. Due to high density and low mean free path in the newborn cloud, the cloud is at least
partially thermalized (Ye et al., 2019; Kočiščák et al., 2020).

2. The electron motion timescale. A portion of the electrons is collected by the spacecraft's body. Simultaneously, a fraction
   of released electrons with energies high enough to surpass the spacecraft's potential well escapes from the vicinity of the
   spacecraft. The former effect does not immediately affect the spacecraft's body potential. The now net positive impact
   cloud remains in near vicinity of the spracecraft's body, counteracting the effect of electron collection by its electrostatic
induction. However, the now net positive cloud is visible as a positive spike in the voltage difference due to its positive
   effect on the antennas, since there is nothing to counteract its influence on the antennas. This effect is asymmetric with
   respect to the 3 channels, since each antenna is influenced differently. The escaping electrons are, however, visible in
   the form of a negative spike, due to their positive impact on the spacecraft's body potential, potentially forming the
   aforementioned negative prespike. This spike is symmetric, since it is due to the change on the spacecraft body. These
two (asymmetric positive and symmetric negative) influences counteract each other and therefore the result is ambiguous,
   depending on the spacecraft's potential, as well as the instrument geometry and impact site, besides other factors.

3. The timescale of the impact cloud retreating from the vicinity of the spacecraft's surface. As the spacecraft body is
   positively charged, the net positive impact cloud is repelled. When the impact cloud's electrostatic induction on the body
   ceases, the electrons previously collected by the body show in the form of a positive peak in the voltage difference,
which we denoted as the primary peak. The rise time of the primary peak is therefore the time that ions need to escape





far enough from the spacecraft body's vicinity, or alternatively, time until the ion cloud is sparse and far enough so that it is shielded by the photoelectron sheath. The peak is in principle the same on all the channels, since it happens on the body, rather than on the individual antennas. An asymmetry might still be visible due to the electrostatic induction of the ion cloud on the antennas that may not have halted yet, discussed in the previous paragraph. This asymmetry halts on a timescale similar to the rise time of the primary peak, as they both depend on ion motion and shielding.

4. The timescale of the impact cloud reaching the antennas. A spike due to ions getting so close to the antennas, that they influence their potential locally. The peak is delayed behind the primary peak due to a finite drift and diffusion velocity of the ions. The antenna charging process is not obvious. Several possibilities for the charging process were previously proposed, observed, and debated. One option is the electric field (Oberc, 1996), possibly due to impact cloud's different potential (Zaslavsky et al., 2012), another option is the charge collection from the impact cloud (Meyer-Vernet et al. (2014); Zaslavsky (2015); Vaverka et al. (2021); Kellogg et al. (2021)). An alternative process of charging due to photoelectron sheath perturbation was also proposed and debated (Pantellini et al., 2012a; Kellogg, 2017).

5. The timescale of potential equalization. Neglecting other influence, the spacecraft's potential is positive and in equilibrium due to balance between photoelectron current with negative dependence on the spacecraft's potential and ambient (solar wind) electron collection current with positive dependence on spacecraft's potential. This balance is perturbed by the net negative charge collection from the dust impact and it is restored on a time scale much slower than the impact cloud motion timescale.

Each phase corresponds to one process being dominant, therefore the phases may or may not begin and end with peaks, which depends on amplitudes and timing for the given event. We note that certain phases may or may not be pronounced in individual waveforms, due to a specific voltage balance, phase timing, or an insufficient temporal resolution of the waveform sampler. Different behavior may be observed in case of a less likely impact onto a scientific instrument, a non-metallic surface, or a non-illuminated back-side of the body. We note that even though the solar panels have a large area compared to the spacecraft's body, they are non-conductive on the front-side, which makes them less sensitive to dust impacts. Much is not understood about the panels' response to the impacts and this is out of scope presently, yet worthy of future investigation.

## 3.1 Charge production equation

The charge is released from the impact site shortly after the dust impact. The amount of charge was found (Auer and Sitte, 1968) to depend on the mass and velocity and is often assumed to follow the empirical equation

$$\frac{Q}{C} = A \left( \frac{m}{\text{kg}} \right)^{\alpha} \left( \frac{v}{\text{km/s}} \right)^{\beta}, \tag{1}$$

where $m$ and $v$ are grain's mass and velocity respectively and $A$, $\alpha$ and $\beta$ are material constants. We note that the process is stochastic and depends on other parameters, such as the angle of incidence of the impact velocity, so the exact charge can not be reliably predicted even if these parameters are known, but Eq. 1 was found to work for the mean charge obtained in





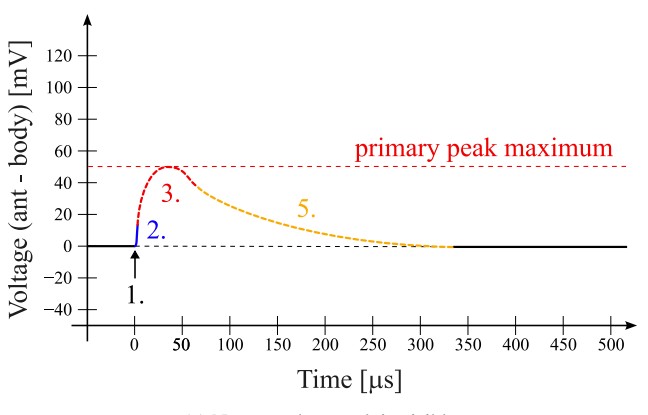

(a) No secondary peak is visible.

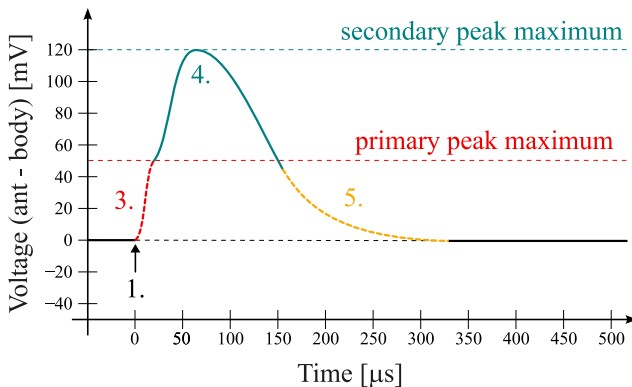

(b) The peaks are discerned by an inflection point.

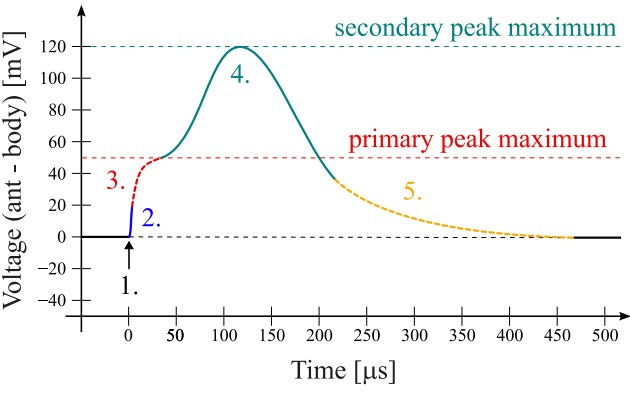

(c) All the phases are clearly visible, although only one local maximum is reached.

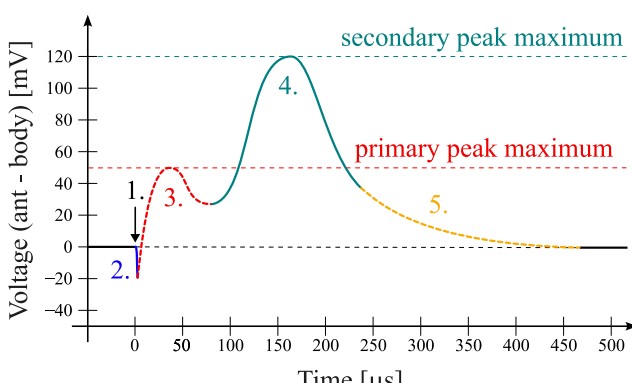

(d) All the phases are visible and two local maxima are reached. The amplitude of the primary peak is 70mV, rather than 50mV.

**Figure 4.** The phases of impact ionization process, as described in Sec. 3. Different eventualities are shown to demonstrate the variability of the pulses that fit the proposed framework. The curves are fictitious, with reasonable primary and secondary peak amplitudes of 50mV and 120mV, as well as a reasonable times-scales. The second phase provides an ambiguous step function and is not otherwise related to a specific shape of the curve.



a repeated experiment. For experimental results and discussion, the reader is referred to Collette et al. (2014) and references therein.

## 3.2 Electron prespike

225 The negative, electron prespike forms due to electrons escaping from the potential well of the positively charged spacecraft. One of the extreme cases is that the potential of the spacecraft is so high compared to the energies of the electrons that virtually no electrons escape and hence, no electron peak is observed. In the other extreme case, the potential of the spacecraft is so low that all the electrons moving initially outward (that is, one half of all the electrons) escape. Since the Solar Orbiter operates in the solar wind and in sunlight, its potential does not usually get below $+5\,\mathrm{V}$, which means that the latter scenario is unlikely. 230 In reality, values between the two extremes are obtained, leaning towards the former scenario.

## 3.3 Primary peak

As the Solar Orbiter is typically positively charged to $\approx 7\,\mathrm{V}$, the positive ions released at the impact are repelled from the spacecraft's body and leave behind the negative charge. It was explained and evaluated before (Zaslavsky et al., 2021) that if the peak is due to a sudden deposition of free charge $Q$ onto the body of the spacecraft, and the antenna's potential $\phi_{ant}$ remains 235 roughly constant throughout the process, the peak's amplitude $V$ is linked to the amount of deposited charge as follows:

$$V \approx \frac{Q\Gamma}{C_{sc}}, \tag{2}$$

where $C_{sc}$ is the electrical capacity of the spacecraft's body ($C_{sc} \approx 355\,\mathrm{pF}$), while $\Gamma$ is the capacitive transfer function between the body and the antenna:

$$\Gamma = \frac{C_{ant}}{C_{ant} + C_{stray}}, \tag{3}$$

240 where $C_{ant}$ is the anatenna's self-capacitance ($C_{ant} \approx 55-70\,\mathrm{pF}$, depending on the variable local plasma conditions) and the $C_{stray}$ is the capacitive coupling between the antenna and the body, including the preamplifier capacitance ($C_{stray} \approx 108\,\mathrm{pF}$). The approximation requires that the rise of the signal is much faster than the relaxation, which is, as we will see, well met. Then we have that $\Gamma \approx 0.34 - 0.39$. Numbers considered, we get that for the primary peak:

$$\frac{V}{Q} \approx 10^9\,\mathrm{V/C}. \tag{4}$$

245 In their recent modelling effort, Racković Babić et al. (2022) concluded that, in the case of STEREO spacecraft with a similar antenna system, the Eq. 2 underestimates the total charge released by about $30\,\%$ due to a finite rise and decay time scales, but is a reliable linear measure of the charge released.

We also note that in case of presence of the electron peak, we evaluate the amplitude $V$ of the primary peak in reference to the low point of the electron peak, that is to the high point of the spacecraft's potential.



### 3.3.1 Antenna induced primary peak asymmetry

First, the electrons are collected from the cloud of the impact plasma. The primary peak then appears as soon as the cloud no longer induces charge on the spacecraft body. This happens gradually, as the cation cloud is drifting away from the impact site and is being shielded by the ambient photoelectrons at the same time. The cloud however influences not only the spacecraft body, but to a lesser extent also each of the three antennas, as debated in Meyer-Vernet et al. (2014). This influence halts on a similar time scale as the influence on the body, but before that happens, this influence is the source of a possible asymmetry of the primary peak as measured with individual channels. Here we will estimate the magnitude of this influence.

Assume a point charge $q$ at the location $\boldsymbol{x_q}$ and the Debye length of $\lambda_D$. The electric potential in the point of space $\boldsymbol{x}$ is then

$$\Phi = \frac{Q}{4\pi\epsilon}\frac{e^{-\frac{|\boldsymbol{x}-\boldsymbol{x_q}|}{\lambda_D}}}{|\boldsymbol{x}-\boldsymbol{x_q}|}. \tag{5}$$

A thin antenna measures a potential of

$$\Phi_{ant} = \frac{1}{|l|}\int_{\boldsymbol{l}} \Phi \, d\boldsymbol{l}. \tag{6}$$

Employing a Monte Carlo model for the charge location on the heat shield, we find that the antennas' response to the point charge near the heat shield is on the same order of magnitude as the response of the spacecraft's body. Therefore the ratio $R$ of primary peak amplitude detected with different channels is often $R \approx 1.5$, see Fig. 5. For more detailed description of the model, refer to Appendix F.

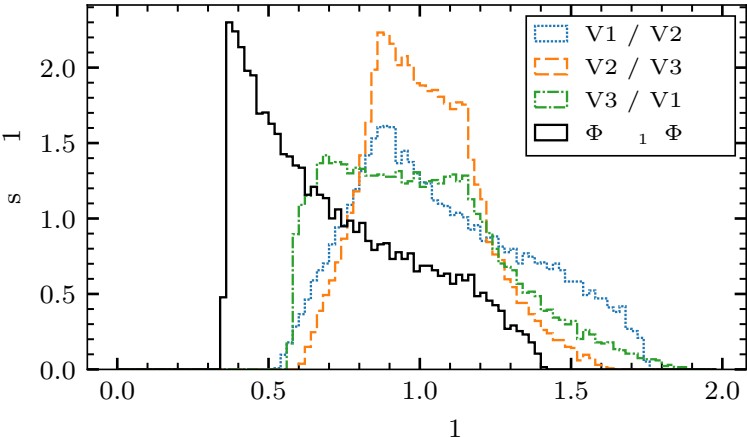

**Figure 5.** The ratio of primary peak amplitudes as predicted by the model for detection in different channels.



### 3.4 Secondary peak

Should the antenna collect free charge from its surroundings, the corresponding voltage would be given by an equation equivalent to the one for the charge collection by the body, but with a different value of the capacitance

$$V \approx \frac{Q\Gamma}{C_{ant}}, \tag{7}$$

270 hence different by a factor of $C_{sc}/C_{ant}$. By substitution for the difference, we find that

$$\frac{V}{Q} \approx 6 \cdot 10^9 \, \text{V/C}. \tag{8}$$

However, the free charge collection is an unlikely scenario and the secondary peak might be caused by various mechanisms. Should the antenna only detect the approaching charge remotely (via induction), its response would depend on the geometry of the encounter: the closer the charge gets to the antenna, the stronger the response, with the maximum equal to the charge 275 collection in case of a very close approach. Should the antenna charge due to photoemission (Pantellini et al., 2012a; Kellogg, 2017), the above mentioned equation holds, but the $Q$ would be the charge due to photoemission rather than to charge collection. Finally, we note that since the secondary peak is decidedly retarded with respect to the primary peak (see Figs. 2, 3), it can not be explained as an electrostatic response to the impact plasma cloud located near the impact site — the motion towards the antenna must be relevant. Besides, we observe the electrostatic response as well, on a different time-scale, in the form of the 280 primary peak asymmetry.

### 3.5 Timescales

The electron peak rises when the electrons no longer induce charge on the spacecraft body. It happens no later than when the electrons are displaced from the spacecraft's body by a displacement comparable to the size of the spacecraft body ($\approx 1 \, \text{m}$). Mind that the energy of the electrons has to be high enough to overcome the positive potential of the spacecraft's body. The 285 temperature of the impact cloud was estimated before (Friichtenicht et al., 1971; Eichhorn, 1976; Collette et al., 2016; Kočiščák et al., 2020) to be $\gtrsim 1 \, \text{eV}/\text{k}_\text{B}$, which implies the electron velocity $v_e \gtrsim 500 \, \text{km/s}$, leading to the rise time of $\tau_e \lesssim 2 \, \mu\text{s}$, which is well below the $250 \, \text{ksps}$ resolution of the sampler, hence it appears instantaneous. If an electron prespike appears stronger on certain antennas, it might indicate that it is partially due to electron collection by the antenna.

Similar to the electron peak, the primary peak appears as soon as the released ions no longer induce their charge onto the 290 spacecraft's body. Two processes cause this: physical displacement of the ions, and the shielding of the ions by the electrons (ambient electrons and photoelectrons). Adopting a moderate ion temperature of $5 \, \text{eV}/\text{k}_\text{B}$ (Collette et al., 2016; Kočiščák et al., 2020) and assuming carbon nuclei, we find the ion speed to be $v_E \approx 9 \, \text{km/s}$. Based on in-situ dust impact measurements, Vaverka et al. (2021) reported $v_E = 27 \pm 5 \, \text{km/s}$ and Racković Babić et al. (2022) reported $13 \, \text{km/s}$. Assuming the average of these two values, $20 \, \text{km/s}$, $1 \, \text{m}$ we find that the displacement happens in $\approx 50 \, \mu\text{s}$ — a time well resolved by the RPW sampler. 295 Should the impact happen within the photoelectron sheath, the photoelectrons are easily the dominant electron population. Assuming a typical $d = 1 \, \text{AU}$ environment, Meyer-Vernet et al. (2017) estimated the timescale for the shielding of $Q = 1.6 \, \text{pC}$





charge to $\tau_{ph} \approx 12\,\mu s$ for an ion speed of $v_E = 10\,\mathrm{km/s}$ and varying as $\propto Q^{1/3}d^{2/3}v_E^{-2/3}$, which is on the edge of the resolution of the RPW sampler.

The potential altered by the net charge left deposited on the spacecraft's body will decay towards the original spacecraft
potential, that is, until the equilibrium is reached again. Under the assumption that the potential perturbation is small compared to the equilibrium potential, the time constant $\tau_{RC}$ of the decay is

$$\tau_{RC} \approx \frac{C_{sc}k_BT_{ph}}{e|I_e|} \approx \frac{C_{sc}k_BT_{ph}}{e^2 n_e v_e S_{sc}}, \tag{9}$$

where $k_BT_{ph}$ is the photoelectron temperature (in eV) and $|I_e|$ is the magnitude of the ambient electron current onto the body of the spacecraft, expanded into the product of the charge, density, velocity and surface $e n_e v_e S_{sc}$. For details, the reader
is referred to Henri et al. (2011). Assuming $C_{sc} = 355\,\mathrm{pF}$, $k_BT_{ph} = 3\,\mathrm{eV}$, $n_e = 5 \cdot 10^6\,\mathrm{m}^{-3}$, $v_e = 500\,\mathrm{km/s}$, and $S_{sc} = 28.4\,\mathrm{m}^2$ we get an order of magnitude estimate of

$$\tau_{RC} \approx 93\,\mu s \tag{10}$$

for typical $r = 1\,\mathrm{AU}$ solar wind environment. It is often reasonable to assume $n_e \propto r^{-2}$.

## 4 Statistical analysis of the primary peak

The primary peaks are found synchronous and with similar amplitude in all 3 channels, therefore we believe that the primary peak is the result of the net charge deposition to the spacecraft's body due to impact. In this section, we examine the statistical properties for the primary peaks, such as the distribution of their amplitudes, rise and decay times. We also compare these to theoretical predictions.

### 4.1 Amplitude distribution

We analyzed the primary peak amplitudes (as described in Section 2.3 and Appendix D) as these are the better measure of the total released charge, compared to the channel global maxima reported previously (Zaslavsky et al., 2021), since the data set now does not contain secondary peaks' amplitudes. The smallest consistently resolved peaks are $\gtrsim 0.5\,\mathrm{mV}$ and the largest included peaks are amplitudes are $\leq 0.3\,\mathrm{V}$. Assuming the relation between the primary peak amplitude $V$ and the charge $Q$ in the form as in Eq. 4, we find the mean charge to be $Q_{mean} \approx 21\,\mathrm{pC}$ and the median to be $Q_{median} \approx 8.1\,\mathrm{pC}$. Further discussion
is available in Appendix E.

The charge production equation (see Eq. 1) for Solar Orbiter is unknown. We assume a production relation as in McBride and McDonnell (1999), that is $\frac{Q}{C} = 0.7 \left(\frac{m}{kg}\right)^{1.02} \left(\frac{v}{km/s}\right)^{3.48}$, and a mean incident velocity as in Kočiščák et al. (2023), $v_{mean} = 63\,\mathrm{km/s}$. We find the mean incident dust mass $m_{mean} \approx 1.5 \cdot 10^{-17}\,\mathrm{kg}$, which corresponds to a spherical dust grain with the diameter of $0.24\,\mu m$, assuming the density of $\rho = 2\,\mathrm{g/cm}^3$.





## 4.2 Rise time


We analyzed the rise time of the primary peak and compared it with the estimates presented in Meyer-Vernet et al. (2017) for the case of sunlit impact surface and for the case of shaded impact surface (see Section 3.5). We adapted the estimates to the median charge of $8.1\,\mathrm{pC}$, as well as the ion speed of $v_E = 20\,\mathrm{km/s}$, obtained as described in Section 3.5. Fig. 6 shows the dependence of the rise time on the heliocentric distance. Inferred means are close to the theoretical (*sunlit*) estimate. The


estimate was done assuming only one (photoelectron) shielding process, while the other (ambient plasma shielding process) as described in Meyer-Vernet et al. (2017) (denoted *shade* in Figs. 6 and 7) plays a role as well. On the experimental side, the exact definition of the rise time is important, as the rise profile is usually not exponential. We define the rise time as the time needed to get from $1/e$ of the maximum to the maximum value of the peak. Fig. 7 shows the dependence of the rise time on the primary peak amplitude, assuming heliocentric distance of $0.75\,\mathrm{AU}$. The data is compatible with the estimate for $V \approx 10\mathrm{mV}$,


but diverges towards either end of the interval. We note that several papers (for example Collette et al. (2016); Nouzák et al. (2020)) suggested that the higher impact velocity might lead to a higher ion velocity $v_E$ in addition to a higher charge yield $Q$, these two effects would then partially counteract each other. The dependence on the heliocentric distance in Fig. 6 is present. Overall, we conclude that the predictions made in Meyer-Vernet et al. (2017) are compatible with the data.

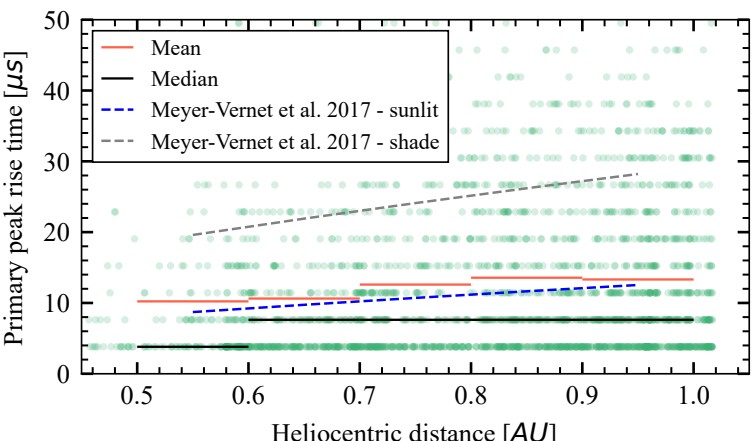

**Figure 6.** Rise times of the primary peaks as a function of the heliocentric distance. Predictions from Meyer-Vernet et al. (2017) are shown in the case that impact cloud shielding is dominated by photoelectrons (sunlit) or solar wind plasma (shade). The predictions are for the median primary peak's charge of $8.1\,\mathrm{pC}$.



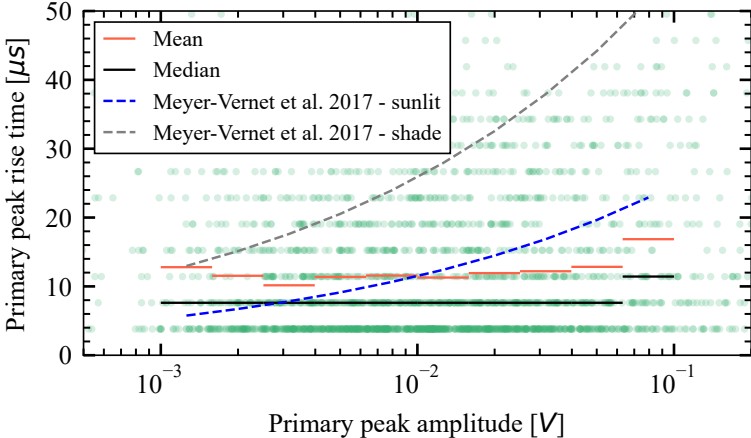

**Figure 7.** Rise times of the primary peaks as a function of the body's peak amplitude. Predictions from Meyer-Vernet et al. (2017) are shown in the case that impact cloud shielding is dominated by photoelectrons (sunlit) or solar wind plasma (shade). The predictions are for the heliocentric distance of $0.75\,\mathrm{AU}$.

### 4.3 Negative prespike

The negative prespike is present intermittently, for example in Fig. 3 (c). The presence indicates that a portion of free electrons was able to escape the spacecraft's potential well. We note that the effect is counteracted by the impact cloud's electrostatic induction on the antennas, which happens simultaneously and on a similar timescale, and which offsets the present effect, possibly to a point when the effect is no longer visible. The exact impact location certainly plays a role, since spacecraft's surface potential is not uniform. On top of that, the spacecraft's potential must play a role as a lower potential implies a

shallower potential well electrons need to overcome in order to escape. To see this dependence, we examine the spacecraft potential data product, based on low frequency receiver measurements of RPW (Maksimovic et al., 2020). We note that this a result of an indirect measurement and therefore the reliability is limited, especially in case of very high or very low values. A correlation between the prespike presence and a relatively lower potential is expected, which is why we show a separate normalized histogram of spacecraft potentials at the times of impacts with prespikes and without, see Fig. 8. Prespikes are

present for nearly any spacecraft potential, but the correlation is apparent, as expected.





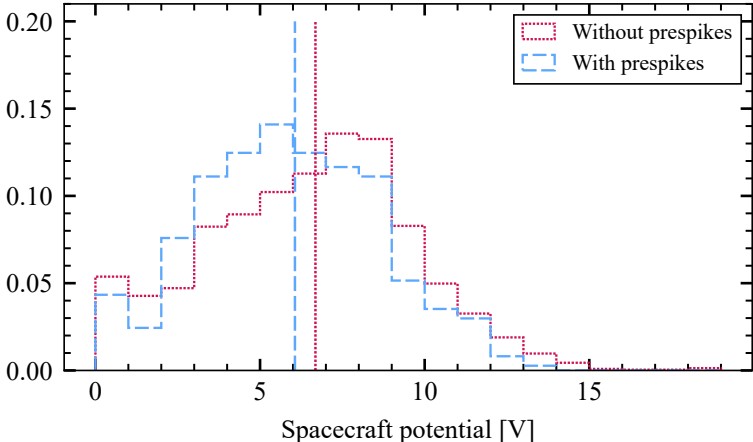

**Figure 8.** The histogram of spacecraft potential at each dust impact for impact with and without prespikes. Averages are shown for the two populations as vertical lines.

## 4.4 Decay time

We established the decay time for the primary peaks as the time to get from $100\%$ to $1/e$, always for the channel that showed the lowest primary peak amplitude, as that is the one least affected by a possible secondary peak. Furthermore, we disregarded any value over $200\,\mu s$. We compare the result to the theoretical values presented in Section 3.5, see Fig. 9. The decay time
shows a clear variation roughly compatible with the model (Eq. 9), albeit with by about a factor of 2 longer close to $0.5\,AU$ and only about $30\%$ longer close to $1AU$. We note that there are uncertainties, for example in the spacecraft capacitance $C_{sc}$ and in the spacecraft surface $S_{sc}$. The shallower dependence might be a result of electron temperature being higher at lower heliocentric distance, which we do not take into account in the theoretical calculation. We also can not exclude an artefact of the secondary peaks that are present, though not visible, as these may introduce error that is hard to estimate.





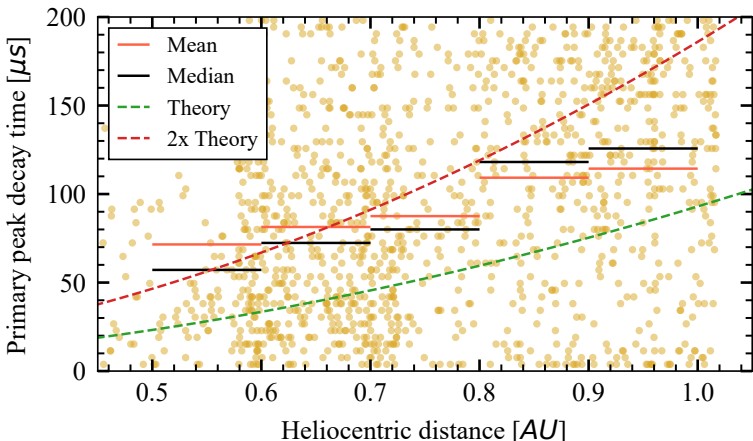

**Figure 9.** Decay times of the primary peaks as a function of the heliocentric distance.

## 4.5 Antenna induced asymmetry


We studied the amplitudes of individual primary peaks in order to compare to the theoretical predictions of Section 3.3.1. We only analyzed the events that show no secondary peak in any channel. In parallel to Fig. 5, ratios of channel pairs are shown in the histogram in Fig. 10, but the information on the ratio between the antenna influence and the body component is clearly not available. The histogram does not show data with the ratio $> 2.2$ and as a result, 5 of 327 values are not shown. Similarly to

the results of the numerical model shown in Fig. 5, values $< 0.5$ are rare, as are the values $\gtrsim 2$, which implies that the process as described in Section 3.3.1 is a good model for the situation, as it explains the magnitude of observed asymmetry.

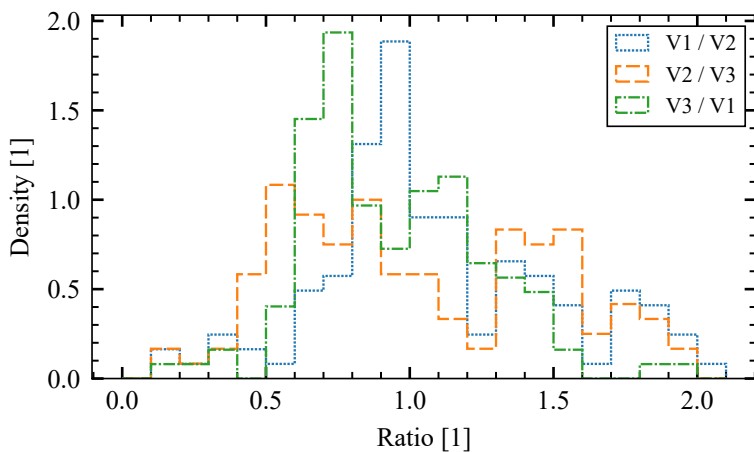

**Figure 10.** Antenna induces asymmetry to the primary peak's amplitude.




# 5 Statistical analysis of the secondary peak

An important proportion of the impacts ($\approx 50\%$) shows a clear double-peak structure, while even more are compatible with the double-peak structure. The secondary peaks' prominent features, where present, include:

- strong asymmetry in the three channels,

- unreliable presence,

- variable, but pronounced delay with respect to the primary peak.

The first point leads to the conclusion that the process causing secondary peaks mainly takes place on the antennas, rather than on the spacecraft body. This implies that, in the process, the affected antenna is charged more positively. The latter two
points imply that the effect relies on a drift of the cations. In this section, we describe statistical properties of the identified secondary peaks.

## 5.1 Delays

The typical delay lays in the range of $100\,\mu s$ to $300\,\mu s$, see the histogram in Fig 11. The secondary peak's delay varies, nearly uncorrelated with the peak's amplitude nor the spacecraft's heliocentric distance, see Figs. 12 and 13. This time is too long to
correspond to charge generation, collection, or even equalization due to ambient plasma currents, as we described all of these earlier and they happen within $\lesssim 150\,\mu s$.

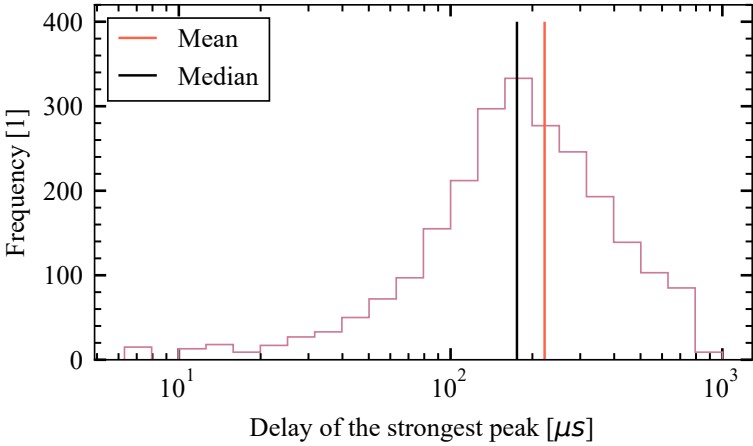

**Figure 11.** Histogram of the strongest secondary peak's delay against the primary peak.

Assuming the ion velocity of $20\,km/s$ as before, the time delay of $100\,\mu s$ to $300\,\mu s$ translates to $2-6\,m$ of displacement. We note that the Solar Orbiter's heat shield's size is approximately $2.4 \times 3.1\,m^2$ and the antennas are $6.5\,m$ long. We therefore conclude that this delay is due to ion motion, since it is the only electric process that happens on this time-scale.



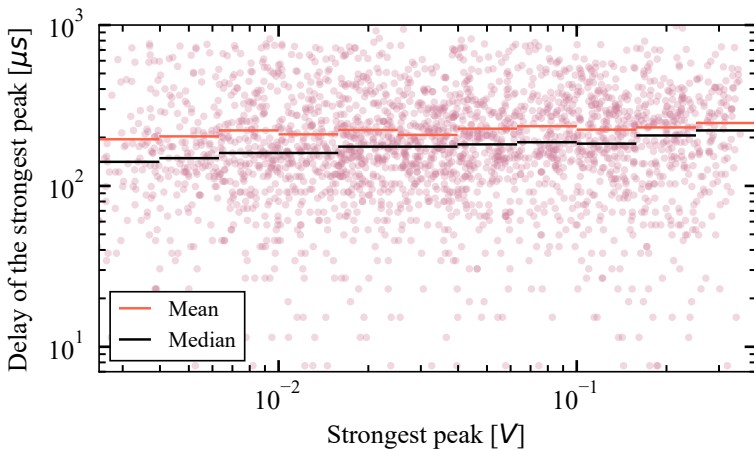

**Figure 12.** Strongest secondary peak's delay against the primary peak as a function of its amplitude.

We note that the delay of $100\,\mu s$ to $300\,\mu s$ is well over the lifetime of the cloud in the photosheath, due to its high electron number density (compare with values shown in Fig. 6 and 7). However, the photosheath decays with the distance from an illuminated surface rather quickly, with the typical Debye length of $0.25\,m$ close to Solar Orbiter's perihelia and $1\,m$ close to $1\,AU$ (Guillemant et al., 2013). We therefore come to a conclusion that a part of the impact cloud often survives the passage through the photosheath to influence the antennas. We also note that the photosheath is not uniform and weaker at places that 390 are less illuminated, such as spacecraft sides.

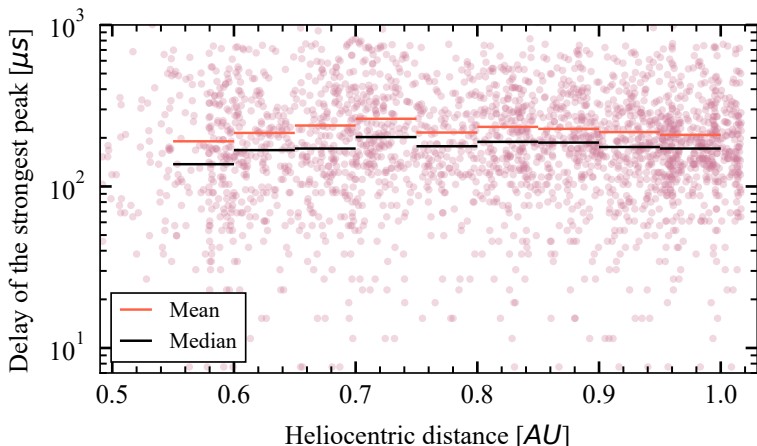

**Figure 13.** Strongest secondary peak's delay against the primary peak as a function of the spacecraft's heliocentric distance.





The delay does not show variation with the peak absolute amplitude (Fig. 12), but it shows a weak correlation with the amplitude relative to the primary peak amplitude, as is shown in Fig. 14. Primary peak's amplitude is a good measure of the total charge released on the impact, and since we study the secondary peak as a random process, normalization to the impact magnitude is natural.

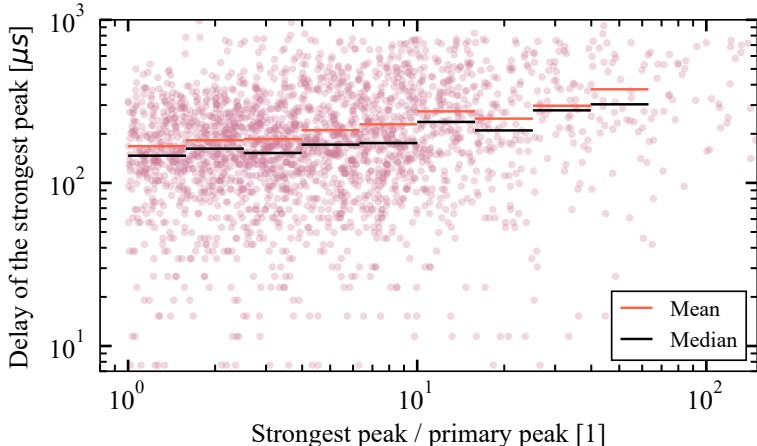

**Figure 14.** Strongest secondary peak's delay against the primary peak on the as a function of he strongest secondary peak's amplitude relative to the primary peak.

We also note that the secondary peak is not only delayed, it also evolves and decays on $\gtrsim 100\,\mu m$ timescale, as is apparent from waveforms shown in Figs. 2 and 3. This hints that the evolution of the secondary peak is also dependent on the dynamics of the ion cloud's motion. This is also consistent with the positive correlation between the secondary peak's relative amplitude and the delay with respect to the primary peak (Fig. 14).

## 5.2 Amplitudes

The secondary peak's amplitude varies and the peak is not always present. We do not claim that small secondary peaks are non-existent, however for the purpose of our analysis, the secondary peaks are considered absent in cases when their amplitudes are much smaller than the primary peak's amplitudes, as then we can not identify them reliably. If the secondary peak is present in a channel, we study its amplitude relative to the amplitude of the primary peak, as the primary peak's amplitude is a good measure of the total charge released on the impact. See Fig. 15 for the plot of relative amplitude of the secondary peak over 405 the primary peak vs. the heliocentric distance in cases where the secondary peak is present. We observe that the typical relative amplitude is between 3 and 5, but often is over 10. There isn't a strong correlation between the relative amplitude of the secondary peak and the heliocentric distance.

Given the time delay that corresponds to the ion motion along Solar Orbiter and what was suggested and observed previously with different spacecraft, one may try to explain the secondary peak as antenna's response to the ion cloud's electric field, be



it via induction (Oberc, 1996; Zaslavsky et al., 2012) or collection (Meyer-Vernet et al., 2014; Zaslavsky, 2015; Vaverka et al., 2021; Kellogg et al., 2021). In the extreme case of the collection of all the created ions by a single antenna, the amplitude would be approximately proportional to the amplitude of the primary peak with the factor of $C_{sc}/C_{ant} \approx 5$. That is ignoring the fact that the ion cloud is exposed to the solar wind and photoelectron sheath environment for $100\,\mu s$ to $300\,\mu s$. The response to the charge colletion is also an upper estimate of the response to the induced fields. We also note that a complete collection of the ions by an antenna is unlikely. The reason is that the antennas present a small cross-section for the ions, since they occupy a small solid angle as seen from usual impact site and are metallic and therefore positively charged. Moreover, we often find the secondary peak in multiple channels, which clearly out-rules the option that one antenna collects all the ions. Therefore the factor of $\approx 5$ is understood as a very safe overestimate of the secondary peak amplitude, if it is due to antenna's response to the ion cloud's electric field. As is shown in Fig. 15, the limit of 5 is breached very often, which implies an amplification of the signal and rules-out the linear response of the antenna to the electric field of the escaping ions. The conclusion is that an amplification process must be present. A similar conclusion was arrived at by (Pantellini et al., 2012b) for STEREO spacecraft's single hits.

The capacitance of the antennas and of the spacecraft increases with decreasing heliocentric distance due to photoelectron sheath's presence, but since a greater portion of the antennas is sunlit compared to the spacecraft body, one would expect a positive correlation in the Fig. 15, should the variable capacitance be important, which is not observed.

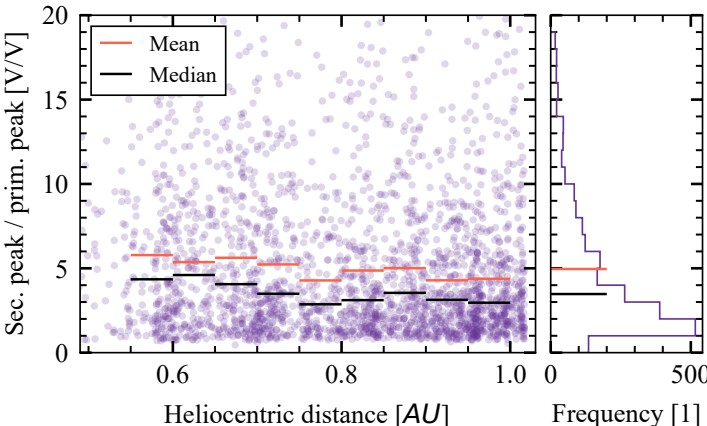

**Figure 15.** The secondary peak relative to the primary peak as a function of heliocentric distance for the events that show a secondary peak. If the secondary peak is present in multiple channels, the strongest one is shown. The absence of values < 1 is due to the secondary peak being obscured by the primary peak. We do not intend to imply there are no small secondary peaks, but we can not identify them reliably.





## 5.3 A possible process

In the Section 5.2 we concluded that an amplification effect must be present near the antennas, allowing none, one, or more of them to be charged beyond the linear electrostatic response to the ion cloud present post-impact.

A mechanism capable of a strong amplification of induced charge detection on thick cylindrical antennas of STERE-
O/WAVES was proposed by Pantellini et al. (2012a) and revised by Pantellini et al. (2013). The idea is that even as the ions do not induce enough response on the antennas, the provided electric field is strong enough to perturb the photoelectron sheath around the antennas, which manifests as a strong transient charging. Pantellini et al. concluded that the effect strength is proportional to the cylindrical antenna's radius, as that is proportional to the photoelectron current. We note that the STERE-
O/WAVES electrical antennas have the diameter of $32\,\mathrm{mm}$ near base (Bale et al., 2008), similar to the ones on Solar Orbiter that have the near-base diameter of $38\,\mathrm{mm}$.

The photoelectron sheath perturbation process as proposed by Pantellini et al. (2012a) is effective once an antenna is partially enveloped by the impact ejecta cloud. Hence, a time delay is expected with respect to the impact on the order of $d/v_{ion}$, where $d$ is the distance from the antenna to the impact site and $v_{ion}$ is the ejecta velocity. We note that this was not observed in case of STEREO single hits (Zaslavsky et al., 2012), but is observed with present results, see Section 5.1.

We perform an order of magnitude estimate of the maximum secondary peak amplitude, assuming that due to envelopment of a portion of an antenna, the photoelectron return current is fully suppressed for a time. A similar estimate was done before by Pantellini et al. (2012a). The secondary peak's amplitude $V_{sec}$ depends on the total charge the antenna accumulates $Q_{ant}$ due to the effect

$$V_{sec} = \frac{\Gamma}{C_{ant}} Q_{ant},$$ (11)

while the accumulated charge depends on the photocurrent density $j_{ph}$, the submerged antenna length $L(t)$, width $w$, and the time $\tau$ during which the return current is suppressed:

$$Q_{ant} = \int_0^\tau j_{ph} w L(t)\, dt.$$ (12)

Assuming a constant photon flux ($j_{ph} = const.$) and a cylindrical antenna ($w = const.$), zero intial expansion ($L(0) = 0$) and a constant expansion speed of the cloud until the maximum expansion $L_{max} = L(\tau)$ is reached in time $\tau$ when the suppression is no longer effective, by integrating Eq. 12, we get

$$Q_{ant} = \frac{1}{2} j_{ph} w L_{max} \tau.$$ (13)

The maximum submerged length $L_{max}$ is related to the total positive charge $Q$ released at the impact, but also to the impact cloud motion geometry, and how much photoelectrons and ambient solar wind electrons are bonded by the post-impact cloud before it reaches the antenna. Again, for the order of magnitude estimate we assume spherical expansion of the impact cloud and neglect the neutralization of the cloud by ambient electrons, therefore the number density $n_{cloud}$ within the cloud of the




charge $Q$ and the radius $L_{max}$ is

$$n_{cloud} = \frac{Q}{e}\frac{3}{4\pi L_{max}^3}, \tag{14}$$

where $e$ is the elementary charge. We note that the fact that the cloud ions are screened by the photoelectrons, does not imply that the photoelectrons remain bonded to the cloud after the cloud has passed the photoelectron sheath — see discussion in Appendix G. Then assuming that the cloud is effective at suppressing the return current until its number density $n_{cloud}$ reaches the solar wind number density $n_{sw}$, we get the radius of the maximum extent of

$$L_{max} = \left(\frac{3Q}{4\pi e n_{sw}}\right)^{\frac{1}{3}}. \tag{15}$$

Then the time $\tau$ to reach this maximum extent, assuming the expansion speed of $v_{ion}$ is

$$\tau = \frac{L_{max}}{v_{ion}}. \tag{16}$$

Considering the Eq. 2 for relating $Q$ and the primary peak amplitude $V_{pr}$, we get the relation between the primary and the secondary peak amplitudes

$$V_{sec} = \frac{\Gamma j_{ph} w}{2 C_{ant} v_{ion}}\left(\frac{3 V_{pr} C_{sc}}{4\pi e n_{sw}\Gamma}\right)^{\frac{2}{3}}. \tag{17}$$

We note that this is a clear overestimate due to the unknown magnitude of the photoelectron screening, besides other uncertainties. Assuming $j_{ph} \approx 6\cdot 10^{-5}\,\mathrm{Am^{-2}}$, $\Gamma \approx 0.37$, $C_{ant} \approx 60\,\mathrm{pF}$, $n_{sw} \approx 10^7\,\mathrm{m^{-3}}$, $w \approx 3.8\,\mathrm{cm}$, and the rest as previously, we get

$$\frac{V_{sec}}{\mathrm{V}} \approx 10\left(\frac{V_{pr}}{\mathrm{V}}\right)^{\frac{2}{3}}, \tag{18}$$

which translates to 100-fold amplification in case of $V_{pr} = 1\,\mathrm{mV}$ and to 21-fold amplification in case of $V_{pr} = 0.1\,\mathrm{V}$. We understand the amplification here as $V_{sec}/V_{pr}$. This is far higher amplification than observed, which is mostly due to the neglect of the photoelectron screening early in the expansion phase in this estimate, as well as the ineffectivity in liberating the photoelectrons from their suborbital trajectories around the antenna. However, a least squares fit of the amplification ratio for the strongest channel (for only the impacts that show a secondary peak) shows a slope of $\approx 0.74$, which is close to the theoretical value of $2/3$, see Fig. 16. Compared to the theoretical estimate, the fit of the amplification is consistent with an additional factor of $\approx 1/10$, which is roughly the product of the portion of impact ions that influence the antennas and the portion of photoelectrons that are liberated, once immersed in the impact cloud. We also note that the fit is influenced by the lower amplitude limit for detection as well as the cutoff at $0.3\mathrm{V}$. We conclude that the Pantellini et al. (2012a) process, as described in present work, provides enough amplification to explain the observed secondary amplitudes.




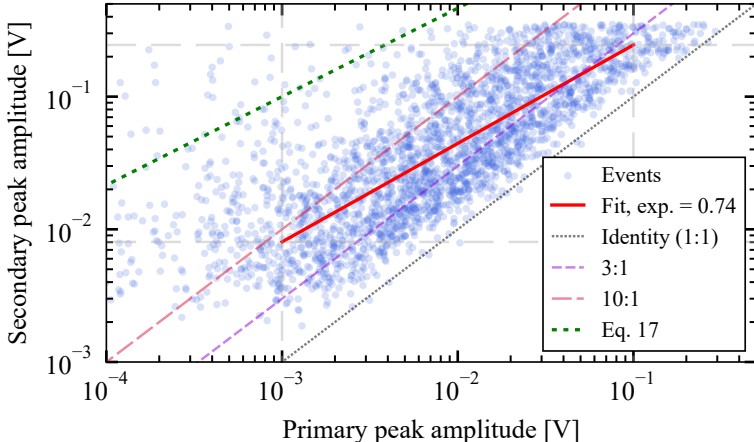

**Figure 16.** The estimate of the amplification factor of the process responsible for the secondary peaks. Each point corresponds to one impact in which a secondary peak was observed. The least squares fit is shown, alongside $1:1$, $3:1$, and $10:1$ linear amplification lines and the Eq. 18.

The ion motion provides a good explanation for the delay of the secondary peak (see discussion in Sec. 5.1), yet the Pantellini et al. process in the present form does not explain the timescale of $\gtrsim 100\,\mu$s at which the secondary peak rises and decays. This is obviously too long for electron motion dynamics, but relatable to ion motion timescales. In the original paper of (Pantellini et al., 2012a), the authors describe how the photoelectron trajectories are temporarily altered due to the presence of a relatively weak electric field of the expanding plasma cloud. This alteration suppresses the photoelectron return current for a time, so that the affected electrons orbit around the influenced antenna's axis. In order to have a longer-lasting secondary peak, as we do, a sink for the excess photoelectrons is required, so that the photoelectrons do not just return to the antenna on the electron motion timescale, which is what is suggested by Kellogg (2017). The claim that the electrons do not return to the antenna they were emitted from is supported by Zaslavsky et al. (2012), who reported the RC decay profile of the pulses that were believed to be caused by the Pantellini et al. process. Since the ion cloud does not provide a field strong enough to liberate a significant portion of the bounded photoelectrons to reach infinity, the sink for the photoelectrons has to be present at around the antenna potential. The only suitable sink here is provided by the spacecraft body. Since the body potential is similar to the antenna potential, the affected electrons orbiting around the antenna axis that are free to migrate along the antenna axis can reach it rather easily. Moreover, due to BIAS subsystem of RPW, the spacecraft body is usually on a somewhat higher potential, compared to the antenna potential (Maksimovic et al., 2020). Given all this, we believe that an important portion of the affected electrons is recollected by the spacecraft body, so the secondary peak is therefore a result of a temporarily amplified current between the affected antenna and the body. A consequence of this is that each such antenna-emitted body-collected electron is counted twice in the affected monopole channel, hence additional amplification. Also, the body potential is changed, albeit





by a difference smaller by the ratio of the antenna's and the body's capacitance, which then shows synchronously in all the channels — a phenomenon that is observed reasonably often.

## 6 Conclusions

We found double peak dust impact signals in about $50\,\%$ of Solar Orbiter /RPW electrical waveforms containing dust impact signatures. To the best of our knowledge, this is the first time such double peak impact signatures were systematically observed.

Upon inspection of the primary peak, we conclude consistence with the state of the art theory for body potential influence. We believe that primary peaks are the better measure of the impact charge, compared to the global maxima of the channel, that are more influenced by the often-present secondary peaks. Our analysis indicates the mean impact charge magnitude of $21\,\mathrm{pC}$ and the median impact charge magnitude of $8\,\mathrm{mV}$. We find that the rise time of the primary peak is variable and consistent with the timescale of the photoelectron sheath shielding of the impact cloud. We find decay time consistent with the timescale

of the potential equalization due to ambient charge collection. We explain the asymmetry between the primary peaks recorded in individual channels with electrostatic influence of antennas, on top of an otherwise symmetric peak caused by the change in body potential.

We observed that the secondary peak is highly variable and very asymmetric with respect to the three channels. A relatively long delay of $\approx 100 - 300\,\mathrm{\mu s}$ with respect to the primary peak suggests that the secondary peak's presence is linked to the

impact cloud moving much closer to the antennas. We conclude that the amplitudes are too strong for either impact charge collection by antennas or antennas being immersed in impact cloud potential. We offer a semi-quantitative explanation, making use of the photoelectron sheath perturbation effect, first described in Pantellini et al. (2012a). Furthermore, we hypothesize that the Pantellini et al. effect might temporarily enhance the current between the antenna and the spacecraft body, as this would explain the longer-lasting nature of the secondary peaks.

*Code and data availability.* The code and the data, including the waveforms for all the identified dust impacts are available on Zenodo, doi.org/10.5281/zenodo.8325050.

## Appendix A: RPW measurement modes

The electrical suite of Radio and Plasma Waves (RPW) consists of three cylindrical antennas. There are three measurement modes: monopole (SE1), dipole (DIFF1) and mixed (XLD1). Whichever the mode RPW is in, it produces three channels of

electrical data. See tab. A1 for the modes' description and Souček et al. (2021) for much more comprehensive explanation.



**Table A1.** The relations between the channels in different measurement modes of RPW. For compactness, $V1; V2; V3$ denote the voltages between the antenna $1; 2; 3$ and the spacecraft body, respectively.

| channel | SE1 | DIFF1 | XLD1 |
|---------|-----|-------|------|
| CH1 | $V1$ | $V1 - V3$ | $V1 - V3$ |
| CH2 | $V2$ | $V2 - V1$ | $V2 - V1$ |
| CH3 | $V3$ | $V3 - V2$ | $V2$ |

Since the device spends by far most time in XLD1 mode, it was chosen as the only mode of interest. Since the monopole data (SE1) are symmetric and the easiest to interpret, the XLD1 data are decomposed to SE1-like data for the analysis and visualization. The decomposition is performed as follows:

$$V1 = CH3 - CH2 \tag{A1}$$

$$V2 = CH3 \tag{A2}$$

$$V3 = CH3 - CH2 - CH1 \tag{A3}$$

**Appendix B: Raw data filtering**

The uncalibrated voltage data `WAVEFORM_DATA_VOLTAGE` of `_rpw-tds-surv-tswf-e_` data is used. Since the data shows a high-frequency artificial modulation at $\approx 80\,\text{kHz}$ and $\approx 110\,\text{kHz}$, the data is filtered with the Butterworth low-pass filter of 32nd order at $f_{lo} = 70\,\text{kHz}$, which leaves us with the temporal resolution of $\tau_{min} \approx 14\,\mu\text{s}$.

According to the system's response function as measured the the RPW's engineering team, there is a significant low-frequency distortion in $< 2\,\text{kHz}$ region. There is also a minor high frequency distortion in $f > 50\,\text{kHz}$ region, which we decided to not correct for, as its impact is very limited. The low frequency part is corrected using Laplace-domain correction, as the very limited window length of $62\,\text{ms}$ introduces other artefacts should the Fourier-domain correction be used. The first order filter with the critical frequency of $f_{hi} = 370\,\text{Hz}$ (see Eq. B1) was found to be the best fit according to the response spectrum, see Fig. B1.

$$v_{corr}(t) = v_{orig}(t) + 2\pi f_{hi} \int_{0}^{t} v(\tau)d\tau \tag{B1}$$





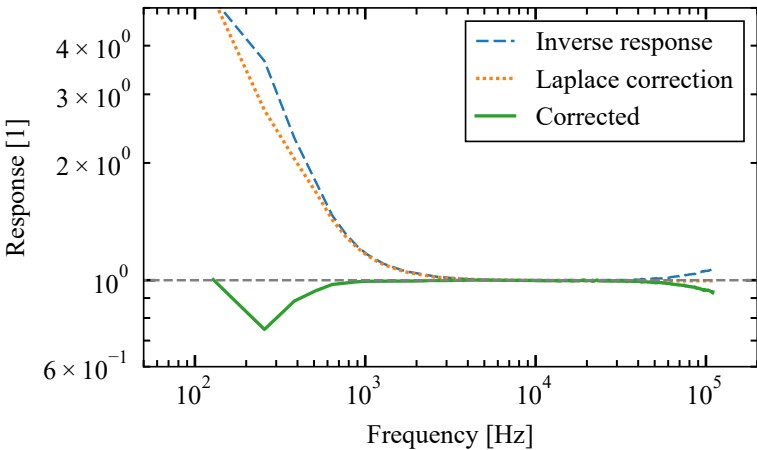

**Figure B1.** The RPW's response function and the Laplace-domain correction

As a result, the corrected signal stays well corrected in the range of $500\,\mathrm{Hz} < f < 70\,\mathrm{kHz}$. We note that higher-order effects
might be present as well, which along with the error we introduce when dividing a small value by another, place a limit on
the reliability of the low frequencies below $500\,\mathrm{Hz}$. For the spectra before and after the corrections, see Fig. B2. For the signal
before and after the corrections applied, see Fig. B3; pay attention to the overshoot attenuated and the secondary overshoot
eliminated.

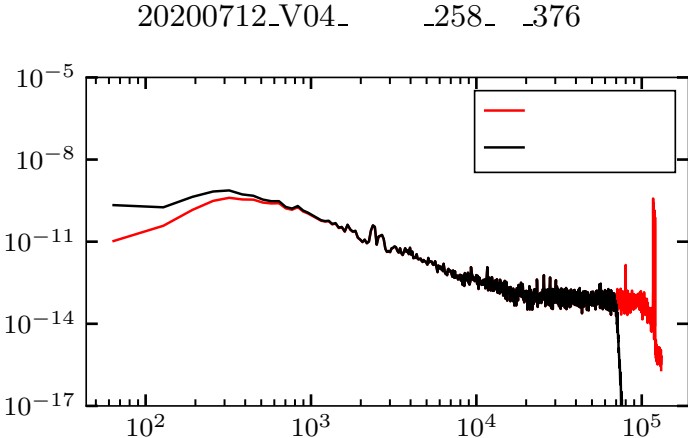

**Figure B2.** The spectrum of an electrical signal, before the low-pass and the Laplace corrections as well as after. We note that Laplace-correction changes the signal on the low-frequency end only, while low-pass filter changes the high-frequency end.



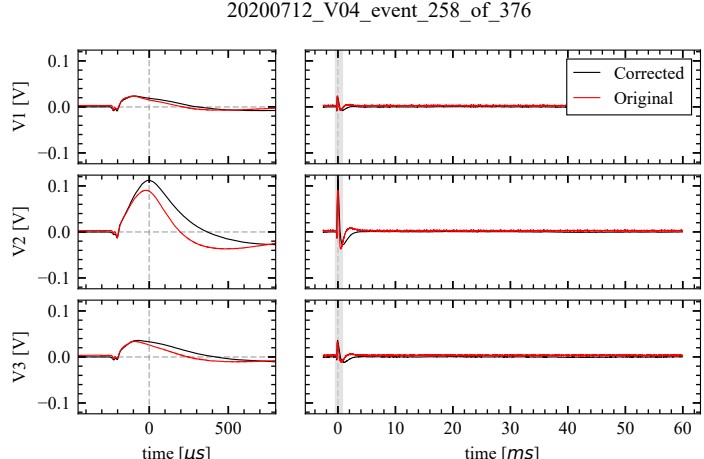

**Figure B3.** The waveform time-series of an electrical signal, the red line shows voltage time-series before the low-pass and the Laplace corrections, while the black line shows the same after the two corrections. The left-hand-side shows detail of the shaded portion of the right-hand-side, which in turn shows the whole recording of 62ms.

## Appendix C: Ternary plot for primary and secondary peaks

The ternary plot in Fig. 1 shows a data point for every event, with the amplitudes based on the channel global maximum. In sections starting with Section 2 we treat the waveforms as containing two major peaks (called primary and secondary), while the latter is not always present. Since we argue that the ternary plot (Fig. 1) shows this indirectly, it makes sense to re-do the ternary plot for the XLD1 events that do and do not contain secondary peaks respectively, see Fig. C1. It is clear that the primary peaks are much more consistent across the channels, compared to the cases when secondary peaks are added.

## Appendix D: Features extraction


The signals of interest (as defined in Section 2.3) were analyzed as follows:

1.  A positive primary peak is assumed to be present in each channel and it is assumed to be of the same amplitude $V_{body}$ in all the channels. The reason is that it is a rather typical case that the primary peak is obscured by a much larger peak in a close succession in at least one of the channels. Therefore, the amplitude of the primary peak is established as the mean

of the amplitude of the weaker two with the reference zero as the mean of the non-affected background signal shortly preceding the impact. The temporal location of the peak is first found approximately using as a minimum of the second derivative near the global signal maximum and then precisely, using a local maximum in the correlation of the signal and a one-sided parabola, which works for both distinct peaks and inflection points. The prespike and body locations are identified as demonstrated in Fig. D1.





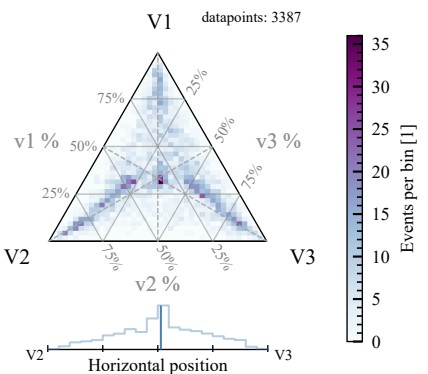 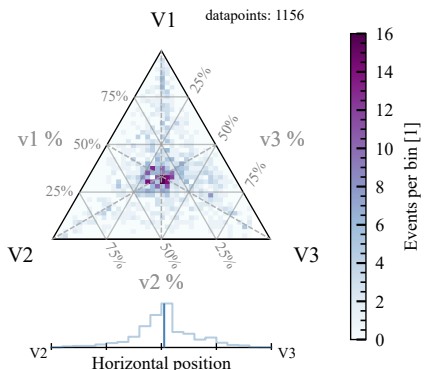

(a) The impacts that do show at a secondary peak in at least one channel.     (b) The impacts that do not show any secondary peak in either channel.

**Figure C1.** Ternary plot for the global maxima of the three monopole channels, one point for each XLD1 event.

2. The rise time of the primary peak is evaluated as the time to get from $43\%$ to $80\%$ of the maximum amplitude, assuming zero on the preceding background level. This range ($37\%$) corresponds to $1/e$ of the maximum and is chosen so that neither the flat nature of the primary peaks nor the background noise influence the estimate.

3. A secondary positive peak may or may not be present in each of the channels separately. First, primary peak is subtracted from the data in the form of asymmetric Gaussian peak with the rise time $\tau_{rise}^{body}$ given by the data and the decay time $\tau_{decay}^{body}$ assumed to be equal to $3\tau_{rise}^{body}$, as that is found to be a good approximation in cases where no secondary peak is present. Second, the secondary peak is considered present if the signal after the subtraction of the primary peak shows a maximum of amplitude of at least $75\%$ of the primary peak. Then amplitudes of the present secondary peaks (after primary peak subtraction) are measured. See this step shown in Fig. D2.

4. The decay time of the primary peak is only evaluated on the channel with the lowest global maximum and is done so as the time in takes the signal to decay from $100\%$ to $63\%$, that is $1/e$. Here we evaluate the decay time closer to the maximum as the undershoot effects and the possible secondary peak influence the result much more than the flat nature of the primary peak or the noise.

5. A negative pre-peak may or may not be present and is assumed to be of the same amplitude in all three channels. The presence is decided by a $3\sigma$ criterion with regard to the noise. If the peak is found present, the amplitude of the primary peak is corrected by this value in the last step.




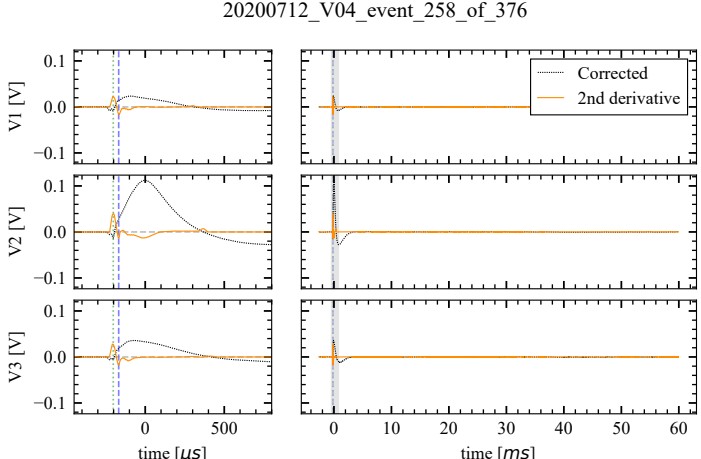

**Figure D1.** The waveform time-series of an electrical signal. The black dotted line shows the voltage signal after the spectral corrections, while the yellow line shows the second derivative. The green and blue vertical dashed lines show the locations of the negative prespike and the primary peak, respectively. The left-hand-side shows detail of the shaded portion of the right-hand-side, which in turn shows the whole recording of 62ms.

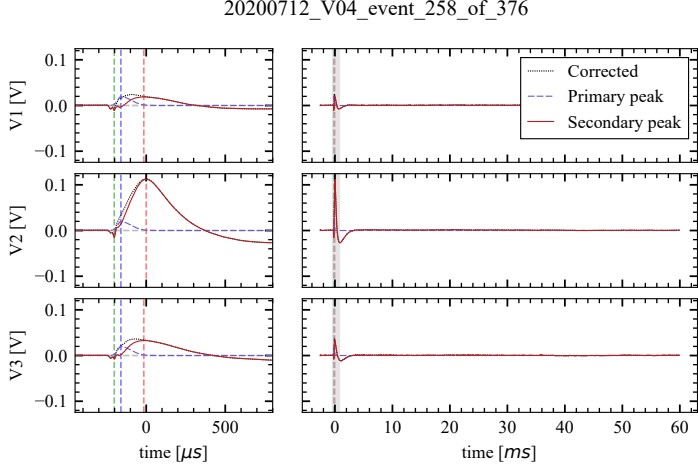

**Figure D2.** The waveform time-series of an electrical signal. The black dotted line shows the voltage signal after the spectral corrections, while the blue dashed line shows the approximated primary peak. The primary peak is subtracted from the measured signal and the residual is plotted as the red line. The green, blue, and red vertical dashed lines show the locations of the negative prespike, the primary peak, and the secondary peaks respectively. The left-hand-side shows detail of the shaded portion of the right-hand-side, which in turn shows the whole recording of 62ms.





**Appendix E: Primary peaks' amplitude distribution**

In Section 4.1 we report on the amplitudes of the primary peaks that are connected to the amount of charge liberated at dust impacts. See fig. E1 for the normalized histogram of the amplitudes. We note that no signals with global maxima over $300\,\mathrm{mV}$ are included, which also disqualifies the signals with $V_{body} < 300\,\mathrm{mV}$ provided that the secondary peak is over the threshold
— leading to underestimation of high amplitude ($\gtrsim 100\,\mathrm{mV}$) counts. Also, given the secondary peak is often the highest amplitude present, recognition of low-amplitude primary peaks is conditioned by the presence of a secondary peak. Therefore, the presence of small primary peaks ($\lesssim 10\,\mathrm{mV}$) is underestimated by a factor that is hard to evaluate. The former bias is more apparent in the black line of Fig. E1, while the latter is more apparent in the light blue line of the same figure.

We note that, contrary to the distribution of global maxima of the signal on an arbitrary monopole (Zaslavsky et al., 2021),
the distribution of the primary peaks' amplitudes does not resemble a power-law. This is not a basis to claim that the power-law is not present in the distribution of amplitudes, or by extension masses, as there is selection bias present, as was mentioned previously.

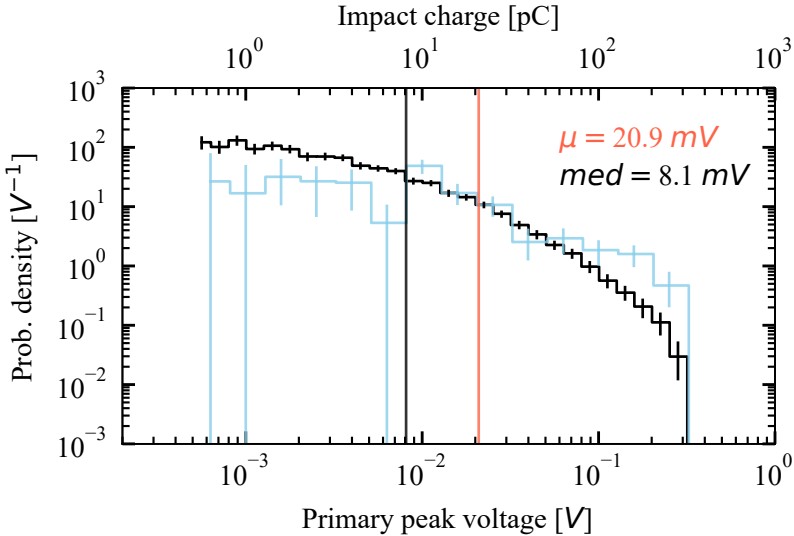

**Figure E1.** Histogram (normalized) of the primary peak amplitudes of all the signals in black, its mean and median are also shown. A separate normalized histogram of only those hits that do not show a secondary peak in any channel is shown in light blue. The vertical error bars represent the $90\%$ confidence intervals obtained by bootstrapping. The conversion from the peak voltage to the impact charge is $V/Q = 10^9\,\mathrm{V/C}$.

**Appendix F: Primary peak asymmetry - the model for antennas' response to a point charge**

The model assumes thin wire $6.5\,\mathrm{m}$ long antennas in a plane. A response of these antennas to a test charge is calculated,
alongside the calculation of the spacecraft's body response to the same charge. In order to produce samples of signal responses,

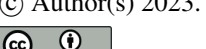


the model samples charge locations (impact spots) from a plane parallel with the antenna plane and $1\,\mathrm{m}$ in front of the antenna plane, in the rectangle of $2.4\,\mathrm{m}$ by $3.1\,\mathrm{m}$, which approximately coincides whit the size and the relative location of the Solar Orbiter's heat shield, see Fig. F1. The potential of an antenna is integrated numerically as the average field along the antenna, according to equations in Section 3.3.1. The value of $\lambda_D$ is assumed $\lambda_D = 10\mathrm{m}$ and is not critical for the results. The `ch1`,

`ch2`, and `ch3` are calculated as the sum of the respective antenna's response with the spacecraft body's response, since the body detects negative, while antennas detect positive charge. We note that a simplification is present: the maxima of the peak of the body response and the peak of the antenna response are typically not synchronous, yet we treat them as such in order to evaluate the ratios of the channel maxima shown in 5.

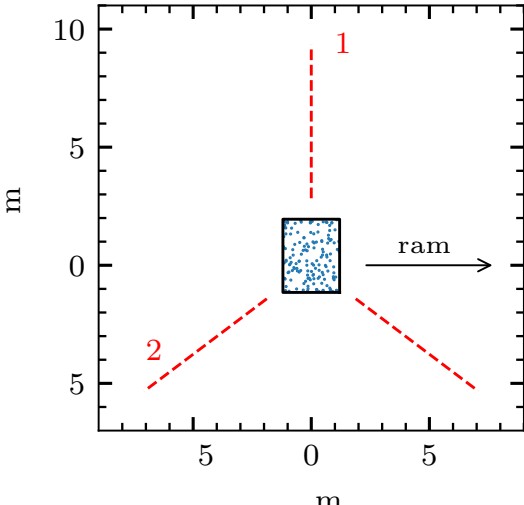

**Figure F1.** The Solar Orbiter's heat shield (black rectangle) and the RPW antennas (red dashed) viewed from behind, as used for the purpose of the antennas' response to a point charge modelling - sampling illustrated.

## Appendix G: Impact cloud potential and photoelectron temperature

The photoelectrons near the illuminated areas of the spacecraft provide a relatively dense ($\approx 10^8\,\mathrm{m}^{-3} = 100\,\mathrm{cm}^{-3}$) region of free negative charges (Meyer-Vernet et al., 2017), with the corresponding photoelectron Debye length of $\lambda_{ph} \approx 0.29 - 0.98\,\mathrm{m}$ at the heliocentric distance of $R = 0.25 - 1\,\mathrm{AU}$ (Guillemant et al., 2013). The photoelectron sheath is therefore effective at screening the escaping positive impact cloud from the spacecraft body after it has passed sufficiently far from the body, which is indeed the process that controls the rise time of the primary peak, see Sec. 4.2 and Meyer-Vernet et al. (2017). However, the

cloud escapes the vicinity of the spacecraft and it is not straightforward to determine, whether it will do so neutralized by the photoelectrons it was exposed to, or not. A possible estimate is done by comparing the typical photoelectron energy with the potential barrier the predominantly positive ion cloud poses for them. Should the photoelectrons be relatively cold, compared





to the depth of the potential hole of the cloud, they are likely to be captured and hence to neutralize the cloud. Should the photoelectrons be much more energetic than the ion cloud potential hole, they are likely to screen only and to not be bounded by the cloud, therefore not neutralizing it.

An order of magnitude photoelectron energy may be done by comparing the incident UV photon energy ($\approx 10\,\mathrm{eV}$) and the spacecraft's surface material work function ($\approx 4\,\mathrm{eV}$), yielding the typical photoelectron energy of $6\,\mathrm{eV}$ near the surface. Guillemant et al. (2013) used the mean photoelectron energy at emission of $3\,\mathrm{eV}$ and $10\,\mathrm{eV}$ in their numerical estimates of the spacecraft charging. The kinetic energy of an electron at its maximum extent from the antenna is very low. Let our order of magnitude estimate be $T_{ph} = 3\,\mathrm{eV}$.

For an order of magnitude estimate of the ion cloud's potential, let us assume spherical expansion of the cloud and a uniform distribution of the charge within the cloud. Assuming the most extreme case, that is the cloud made of cations only, the mean charge of the cloud is $Q \approx 21\,\mathrm{pC}$ (see Sec. 4.1). Then the potential $\Phi$ within the cloud of radius $R$ at the distance from the center of $r$ is readily obtained as

$$\Phi = \frac{1}{4\pi\epsilon_0}\frac{r^2 Q}{R^3}. \tag{G1}$$

The maximum potential is present at the edge of the cloud ($r = R$), that is

$$\Phi_{max} = \frac{1}{4\pi\epsilon_0}\frac{Q}{R}, \tag{G2}$$

which numerically is

$$\Phi_{max} \approx \frac{0.2\,\mathrm{Vm}}{R}, \tag{G3}$$

or

$$\Phi_{max}(R = 10\mathrm{cm}) \approx 2\,\mathrm{V}; \tag{G4}$$
$$\Phi_{max}(R = 1\mathrm{m}) \approx 0.2\,\mathrm{V}. \tag{G5}$$

We see that the simple order of magnitude estimate shows that the potential within the impact cloud drops below the photoelectron energy well within $10\,\mathrm{cm}$ of expansion, suggesting that one may neglect it in calculating the photoelectron current collected by the cloud.

*Author contributions.* Concept: SK, AZ, JV, AT. Data analysis: SK, AZ. Interpretation: SK, AZ, NMV, JV, AT, IM. Manuscript preparation: SK.

*Competing interests.* One of the authors is a member of the editorial board of Annales Geophysicae.



*Acknowledgements.* The work was partially done during SK's stay with OBSPM, LESIA and SK thanks them for their hospitality. Authors
sincerely appreciate the support of Solar Orbiter/RPW Investigation team. This work made use of publicly available data provided by A.
Kvammen at Zenodo, doi.org/10.5281/zenodo.7404457. AT and SK were supported by the Tromsø Research Foundation under the grant
19-SG-AT. IM was supported by the Research Council of Norway under the grants 262941 and 275503. JV was supported by the Czech
Science Foundation under the grant 22-10775S.



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
