# Peer review of "Impact Ionization Double Peaks Analyzed in High Temporal Resolution on Solar Orbiter"

_EGUsphere, 2023_

## Author Response (AR1)

Dear Zoltan,

thank you for your feedback. You will find our responses to your comments in the end of theis post. Your comments were incorporated into a revision-in-progress as indicated in the response. We sincerely hope that our response proves satisfactory to you, and we encourage you to inquire further should any additional questions arise.
* * *
RC: Section 1.1, around line 50. The manuscript provide the surface area of the SC that is sensitive to dust impacts. For dust flux calculations, the effective cross section the SC would be more relevant.

AC: Agreed, the cross-section as seen from the Sun and as seen from the ram direction were added.
* * *
RC: Section 1 would really benefit from presenting a figure of the SC and the arrangements of the antennas with the indication of the ram direction.

AC: Agreed, added.
* * *
RC: Section 2.2 presents the bandwidth of the electronics but the text does not specify whether the presented waveforms were or were not corrected for effect of the electronics. This would be important to clarify as some features in the figure and some of the values from the analysis could be affected by the finite bandwidth.

AC: All the shown waveforms are corrected as best we can. The text was clarified.
* * *
RC: Section 3, lines around 161. The manuscript provides a list of references that the general understanding of the antenna signal waveform is based on. This list, however, is missing some of the fundamental references, namely Collette et al. (2015) , Nouzak et al. (2018), and Shen et al. (2021 a AND 2021b). The experimental study papers were instrumental to recognizing the physical principles responsible for some of the characteristic features in the dust impact waveforms (e.g. the interpretation of the prespike, or demonstrating the importance of induced charging). These articlesvery much belong to the list.

AC: Agreed, those are relevant references. The references were added to the list.
* * *
RC: Section 3, bullet point (2), line 180. "However, the now net positive cloud is visible as a positive spike in the voltage difference due to its positive effect on the antennas, since there is nothing to counteract its influence on the antennas. " This sentence is difficult to follow. Is it talking about the electron escape and the prespike? If so, the measured spike should be negative. Please revise for more clarity.

AC: The bullet point (2) was clumsy. There are two fast spikes: 1. electron escape prespike in the body potential and 2. electrostatic detection of the ion cloud with antennas. These happen on a similar time scale and having opposite poloratiy, they (partially) counteract each other.
* * *
RC: Section 3, bullet point (4), lines around 200. The reference list is again omitting the very relevant publications by Shen et al. (2021b) and Shen et al. (2023). These experimental studies clearly demonstrated that the dominant process responsible for generating signals on the antennas is due to induced charging from the expanding ion cloud. This needs to be discussed in this section and the references properly cited.

AC: Agreed on the point that, usually, the dominant process causing the dipole signal is the induced charging of the antenna. However, induced charging peak rises (as clearly shown in row2 col1 of Fig. 2 In Shen et al. 2023) on a much faster time scale (purple, <5us in Shen's plot), compared to the time scale of ions retreating from the vicinity of the spacecraft (blue, orange, ~30us in the same plot). This is in fact one of the two fast effects discussed in the bullet point 2 (the other is the effect of escaping electrons, causing the neg. prespike). This effect should not be confused with what is presented here in the bullet point 4, i.e. the slow asymmetric peak observed on a time scale of >100us, i.e. far too slow for electron motion time-scale and therefore not observed by Shen 2021b nor or 2023. The peak this bullet point aims to describe is the peak at ~500us in Fig. 3 panel d of the present paper and the process captured by the 4th phase in Fig. 4 of the present paper. The presence of this peak and the

fact that it is not consistent with the effect of induction of the ion cloud on the antennas is one of the key points of the present paper. The suggested discussion was added for more clarity. Also, this place is after all not found appropriate for this discussion, therefore the discussion of different processes possibly causing the peak is left for later, i.e. section 5.2.

—————————

RC: Section 3.3, equations (2) and (3) and text around it. It should be noted that this is only a simplified representation of how the measured voltage is related to the charge and that the full treatment was recently published by Shen et al. (2021b) that includes the effect from all antennas and the SC body.

AC: Agreed, the reference was added.

—————————

RC: Section 3.3.1. The first sentence ("First, the electrons are collected from the cloud of the impact plasma. The primary peak then appears as soon as the cloud no longer induces charge on the spacecraft body. ") is misleading and/or incorrect. Should be revised for more clarity

AC: The text was misleading and was revised for more clarity. The misunderstanding probably stems from us making a distinction between the primary peak (change of the potential of the body, ignoring the antennas) and the ions' influence on the antennas, while e.g. Shen et al. 2021b and Rackovic-Babic et al. 2022 treat these as merely two terms in the matrix product. It of course makes sense to treat them as such as they do, however, we focus on the time-scales of the individual processes and the influence fo the antennas happens much earlier than the influence of the body potential due to ion escape. Our point here is that the primary peak rises slower than the influence of the antennas, the latter of the two is the subject of eqs. 5,6 and this subsection as a whole. Our approach of making the distinction between the effects is somewhat similar to what is shown in Fig. 2 of Shen et al. 2023 with a clear distinction between body potential (ants 1, 2) and the antenna response (dipole 3-4) rather than with e.g. Fig. 4 of Shen et al. 2021b, which shows several differently influenced monopoles.

—————————

RC: Section 3.3.1 line 253. This is a comment so no change is required. In my opinion the photoelectron cloud cannot 'shield' the ion cloud from the impact plasma. The photoelectrons are a zero sum to the charge balance to the system and thus cannot interfere with the effect of the ion cloud. The photoelectrons (dominant majority) is energetically tied to the SC and their generate a

permanently present induced charge on the spacecraft that is essentially the same as if they were ON the spacecraft. Thus any simple rearrangement of the photoelectrons cannot lead to shielding or changing the charge balance of the system as a whole.

AC: Agreed that the rearrangement of the photoelectrons does not change the net charge in the system, but shielding of the ion cloud may happen, altering the trajectories (and hence the spatial distribution) of the bound photoelectrons which does not require the change in the charge balance in the whole system, yet will mitigate the influence of the ion cloud on the potential of the body.The ions are not bound to the spacecraft and as they escape thorugh the photoelectron sheath, the induction on the spacecraft can not be the same as if there was nothing between the ions and the body, but vacuum.
* * *
RC: Section 3.3.1, equation (5). This equations needs some explanation. Is this equation applied to the SC, the antenna, or both? This should be clarified and a reference given. I will also comment that the exponential Debye shielding parameter not relevant in collisionless plasmas, such as the solar wind. Without collisions, there is no trapping mechanism for an electron cloud to form around the positive impact charge cloud. Here is one reference, for example: https://www.proquest.com/docview/304427039/fulltextPDF/5FB99845C03A4CA5PQ/1?accountid=14503

AC: Eq. 5 is basic and general. Eq. 6 is the application to the case of a thin antenna and it is the only application of eq. 5 here. The Phi_body in Fig. 5 is assumed to be equal to the ion cloud's charge itself. The clarification as to what is the Phi_body was added in app. F. The effectivity of shielding is not clear, however it is not crucial either and ommitting it altogether would not change all that much. It is not clear to what extent the photoelectrons shield a positive charge, but it definitely does not change the character of the process. I can't agree that there is no shielding in collisionless plasma. The reference that you provided claims that shielding is going on, albeit with a different profile, equation, etc. In re current revision, we ommitted Debye factor here for simplicity and this does not make much difference.
* * *
RC: Section 3.3.1, equation (5). This would be a good place to refer to the work by Shen et al. (2021b) that provides the full equation for calculating the potentials on the elements (SC and antennas) with the full consideration of the geometry effects. This is important as with multiple elements the charge induction is split between them.

AC: The reference was added, as well as the conclusion that our results are in agreement with the results of Shen et al. 2021b.
* * *
RC: Section 3.4, equation (7) and text around it. The sentence above the equation is phrased such a way that it may suggest that charge collection is needed for equation 7 to be valid. This is not the case and the cumulative induced charge from the ion cloud should have the same effect. Again a good place to refer to Shen et al. (2021b) that has presented the full equation for the signals charging mechanisms.

AC: The sentence was rephrased for more clarity. One of the points of the paragraphis that the signal is not consistent with the induction from the ion cloud due to the great delay. Therefore the effect of interest was not studied by Shen et al. 2021b who observed no such delay of >100us.
* * *
RC: Section 3.4, line 272. "However, the free charge collection is an unlikely scenario". This is indeed true. O'Shea et al. (2017) pointed out that only a tiny fraction of ions from the impact plasma could be collected by the antennas. This reference is relevant here and should be cited.

AC: Thank you for the reference, added.
* * *
RC: Section 3.4. line 277. 'decidedly retarded' - not clear what this means. A revision is suggested.

AC: Rephrased for more clarity as "noticeably retarded by >~ 100 us"
* * *
RC: Section 3.5, line 292. Shen et al. (2021a) have directly measured the ion speeds in the laboratory for a reduced sized model spacecraft. This work is relevant and useful here and should be properly referenced, i.e. not just mentioned, but discussed what was done and what results it yielded.

AC: Agreed, thank you for the reference. The reference mentioning the methods and the findings was added here.

—————————

RC: Section 4.2, line 335. The Shen et al. (2021b) paper is again rather relevant here and should be included in the reference list.

AC: If my understanding is correct, Shen 2021b in fact did not observe the correlation between a high impact speed and a high ion speed, since for the impacts in their fig. 6 and fig. 7 the dependence seems inverse, if anything. The reference was added with a note that the correlation was not observed. Let us not forget that the impact velocities in situ are typically >50km/s, while the accelerator results are typically at <30 km/s, besides other important limitations of ground-based measurements.

—————————

RC: Section 4.2, line 337. 'these two effects would then partially counteract each other. ' It is not clear what effects are counteracting and a revision is advised.

AC: The text was revised for more clarity.

—————————

RC: Section 4.2. A general comment here. This section is written such that it supports the theories predicting the variation of the risetime with heliocentric distance and amplitude (Figs 6 and 7). The results in these figures instead show that there is significantly less variation than expected. (1) I think the more realistic description here should be that only a small variation is observed with heliocentric distance and the authors are encouraged to be open to the possibility that the theory is not supported. As I argued above, photoelectrons are not capable of shielding and the data appears to be in clear agreement with this. (2) I would also point out that the antennas on the parker solar probe have observed the long lasting charged signals from debris released form the spacecraft. There was no shielding by photoelectrons present. (3) I can offer an alternative explanation for the somewhat lower peak rise times  at 0.5 AU. As Shen et al. (2021b) demonstrated, the 'risetime' of the primary peak may appear faster if the ion cloud is expanding over the antenna. Would it be possible that with decreasing heliosphere distance more impacts occur in the ram direction, and thus closer to the corresponding antenna? (4) And lastly, Figure 7 seems to directly contradict the predictions from work of Meyer-Vernet (2017). This should be addressed instead of making the unsubstantiated

statement that "Overall, we conclude that the predictions made in Meyer-Vernet et al. (2017) are compatible with the data. "

AC: I agree that the claim that the fit is good is an overstatement, and this was addressed in the revision. I do not agree that fig 6 shows significantly less variation than predicted. The model is simplified in that it assumes a constast solar wind, a contant VUV irradiation (both in time and the location of the impact), a constant ion speed, besides other simplifications. Since these in reality vary a lot, the mean risetime is shown and variation with heliocentric distnace is rather clearly present. I agree that even the model that does not rely on photoelectrons shows the same variation (dashed gray), however the values just don't fit with these, as only a small portion of riestimes is higher than 20 us and as you mentioned, and there are definitely some impacts into the shadow, which explains these data points. We acknowledge the contribution of the antenna-induced component, which has a mush faster rise time and this was added to the text for clarity. However, we have reasons to believe that this contribution is not critical: 1. the channel which shows the smallest primary peak is analyzed, to avoid analyzing one that might be severly influenced by ions 2. the definition of the risetime as the time to get from 1/e of the maximum to the maximum is immue to whatever happens in the first 1/e of the rise and is more sensitive to the end of the rise, just before the maximum, therefore to the slower of the two components, and 3. the fast, induction, component is rarely equal in amplitude to the body-charge component, therefore it surely does not span the factor of 2-3 needed for the points to overlay with the gray dotted line. Let me note that PSP is on much lower potential, since the events were mostly recorded at < 0.2AU. Moreover, the antennas of PSP are 3mm thin and therefore provide comparitively low photoelectron current. There is definitely a thick photoelectron sheath around solar orbiter's heat shield and an important sheath around its 3cm tihick antennas. The point with more impacts close to one of the antennas at lower heliocentric distances is very interesting and we considered this before, yet it does not show in the data: the amplitude balance on the three channels doesn't seems to vary with the heliocentric distance much, which implies that the distribution of the impact locations on the spacecraft's body does not change singificantly with the heliocentric distance. We belive this is because Solar orbiter's azimuthal velocity changes by less than a factor of 2 between 0.5AU and 1AU and the velocity of beta meteoroids is also higher at lower heliocentric distances, as shown in https://doi.org/10.1051/0004-6361/202245165. This argument is perhaps beyond the scope of the section. We agree that fig 7 shows significnatly less variation than expected and that part was rephrased for more transparency. The main point here is that the risetimes show better compatibility with the "sulit" alternative than with the "shade" alternative.
* * *
RC: Section 4.3, line 341. Not clear what this sentence means: "We note that the effect is counteracted by the impact cloud's electrostatic induction on the antennas, which happens simultaneously and on a similar timescale, and which offsets the present effect, possibly to a point when the effect is no longer visible. " The prespike mechanism was accurately described earlier in the manuscript, but this is confusing. Please revise for more clarity.

AC: The electron prespike was described theoretically and very briefly near figures 2,3,4. In section 4.3 we inspect the prespike in the data. The unclear sentence was rephrased for more clarity. The main point is that the prespike and the induction on the antennas happen on a similar time scale, which explains why the amplitudes of the prispike in different channels might be different.
* * *
RC: Section 4.4. Line 353. "Furthermore, we disregarded any value over 200µs. " It is not clear how is this limitation justified. By looking at figure 3, the decay times appear to be just about 200 microsecond, or maybe even longer. What is justifying this limitation?

AC: There are many events that show a secondary peak, and this is important for the manuscript. Since we have a decently proven estimate of the range of the decay times which worked many times before for diferent spacecraft, we only focus on this range to get as little contamination by merged double peaks as possible, as well as to avoid extreme outliers. The value of 200us was chosen as double the value expected at 1AU. Changing the limit to 300us changes very little, showing similar factor of 2, yet moving the means and medians a little higher. The only important conclusion here is that the primary peak here is what is usually considered the "monopole response" to an impact. You are however right that is seems from fig. 3 that there are decay times around or above 200us. The longer decay times come from the charging of the antennas, compared to the shorter decay times that come from charging of the body, since their RC constant is differend, due to the area and geometry of the antennas compared tp the sc body. This difference is observed in Fig. 3 if you look the the primary peaks in panels b,c,d as compared to the secondary peaks of these panels or the panel a.
* * *
RC: Section 4.4, line 354. "The decay time shows a clear variation roughly compatible with the model (Eq. 9)." This is another statement that is not substantiated by the data presented in Fig. 9. The variation should be a factor of 4, but it is only by a factor of 2. Please revise the statement for accuracy.

AC: Agreed that the statemant was exaggerated. It was rephrased to be more on point. The estimate is an order-of-magnitude one and you are right that 2*Theory is not a great fit either. Therefore in the current revision we rather show Theory+35us, which makes for a better fit. The constant offset may be an effect of the defintion of the decay time as the time to get from max to max/e, while the shape, especially near the maximum, is not exponential. Mind that the assumptions on the dependence of plasma density, plasma components energy distribution, and vuv illumination as functions of the heliocentric distance are simplifications and in reality they vary a lot, which also has an impact on the somewhat lower factor observed. A detailed explanation is not the point of this paper and is worthy of future investigation.
* * *
RC: Section 5, line 371. What does 'unreliable presence' means? Please revise.

AC: Agreed that the sentence made no sense as "features where present: ... - unreliable presence". They point is that the peak is observed in some (many) of the wavefroms but not in every one. The text was revised for more clarity.
* * *
RC: Section 5.1, line 282. The manuscript assumes 20 km/s ion velocity, which is a bit high to properly interpret the 100-300 us time delays considering the antenna lengths. The experimentally measured values by Shen et al. (2021a) were closer to 10 km/s. This latter value would give a better agreement with the data and may be worth mentioning.

AC: Sure, if we say 10km/s, the conclusions don't change. Shen 2021a was added as a reference where the ion velocity is discussed in sec. 3.5 and I agree it is relevant. As it stands, accelerator (the most replicable) results show 8-12 km/s, STEREO (probably the closest analogue) shows 13km/s and MMS (the most basic measurement, using tip-sensitive antennas) shows 27km/s. WE believe it is perhaps best to show estimates following both numbers, i.e. 10 and 20 km/s wherever relevant.
* * *
RC: Section 5.1, lines 385 and figure 13. The manuscript states that the 100-300 us are well over the lifetime of the cloud in the photosheet and the figure demonstrates little variation with heliocentric distance. These are both clear evidences that the photoelectron sheet has little effect on the interaction between the impact plasma cloud and the SC-antenna system. (Note my comments from above, including the one on the PSP). Cumulatively, this manuscript presents all the evidence needed to conclude that the photoelectron sheet theory does not agree with the observations.

AC: Since the plasma is collisionles and as you pointed out, the photoelectrons do not get attached to the ion cloud, hence the usage of the word "lifetime" was wrong. The conclusion that photoelectrons do not get attached to ion cloud is also supported by the estimates in app. G. The fact that delay does not depend on heliocentric distance nor amplitude just shows that the ion expansion speed is fairly constant, or at least does not vary with these two parameters. The text was revised to not be misleading.
* * *
RC: Section 5.1, line 388.'We therefore come to a conclusion that a part of the impact cloud often survives the passage through the photosheath to influence the antennas.' Not clear what this means.

AC: The text was revised for more clarity. The point is that at least part of the impact ejecta will reach the antennas.
* * *
RC: Section 5.2, line 415. "The reason is that the antennas present a small cross-section for the ions, since they occupy a small solid angle as seen from usual impact site and are metallic and therefore positively charged." This is exactly the finding of the work by O'Shea et al. (2017) and it should be noted here.

AC: Thank you for the reference, it was added.
* * *
RC: Section 5.2, line 420. I find the conclusion that necessitates the presence of some amplification process more speculative than substantiated by the data and arguments presented. I will point out that the large (Sec peak) / (Prim peak) ratio could be simple explained by the primary peak amplitude being smaller than expected. The primary peak amplitude is simply given by the difference between the collected fraction of electrons and ion from the impact plasma (see for example Shen et al. (2021b) for more details). If the same amount of electrons and ions are collected by the spacecraft, the primary peak amplitude goes to zero, seemingly increasing the value of the (Sec peak) / (Prim peak) ratio. All that needs to happen are changing condition of the impact plasma or SC potential to explain the observed values.

AC: We have of course considered the possibility of the primary peak being too small rather than secondary peak being large. The situation when a similar number of electrons and ions is collected is feasible at low spacecraft potential (potential < impact cloud temp.) but very unlikely at higher spacecraft potential, such as 10V. If this was the reason for the high sec/prim ratio, one would expect high ratios mostly at lower potentials. However, the spacecraft potential does not seem to predict the ratio: the mean sec/prim ratio at scpot around 0V is 5.27 while at scpot around 8V it is 4.83. The relation between the amplification and the spacecraft potential is very similar to that shown in fig 15, as potential and heliocentric distance are rather well correlated. We do not claim to know for sure what causes the process, but the ratio of 5 is breached nearly half of the times even while scpot is relatively high. Besides, the electrostatic induction is observed as its own effect and is far too fast to

cause this, the effect in question must be more local, as the time delay between the priamry and the secondary peak implies.

——————

RC: Section 6, Line 506. The conclusion should discuss the oblivious which is that the secondary peak may inform on the impact location and one could learn from the source (orbital parameters) of the dust particle generating the signal. This has been also pointed out by Shen et al. (2021b).

AC: Agreed, the sentence was added: Importantly, the amplitudes of the secondary peaks are likely related to the impact location on the spacecraft, which is worthy of future investigation.

——————

Dear Referee,

thank you for your feedback. You will find our responses to your comments in the end of theis post. Your comments were incorporated into a revision-in-progress as indicated in the response. We sincerely hope that our response proves satisfactory to you, and we encourage you to inquire further should any additional questions arise.
* * *
RC: Consider presenting an illustration of the SC and the antenna arrangement. It can be included in Section 1 or Appendix A. The inclusion of such an image will allow a more complete understanding of the geometric relationship between the spacecraft and its antennas.

AC: Agreed, the sketch was added in Sec. 1.
* * *
RC: Section 2.3. (Line 154.) "Besides, we disregarded the signals captured very near the beginning or the end of the recording window." It would be helpful if you could explain the reasons for this criterion. You should also specify the parameters that define the time limit that is considered 'very close' to the beginning or end of the recording window.

AC: The reason and the exact criterion were added in the text: "Besides, for the sake of data quality, we disregarded the signals captured very near the beginning or the end of the recording window, that is within the first or the last $100$ samples, or $0.38 \unit{ms}$, since these often do not show the full peaks of interest."
* * *
RC: Section 4.3. (Line 341.) "We note that the effect is counteracted by the impact cloud's electrostatic induction on the antennas, which happens simultaneously and on a similar timescale, and which offsets the present effect, possibly to a point when the effect is no longer visible." Something is missing in this sentence! Please change it to make it clearer.

AC: The section was revised for more clarity.
* * *
RC: Section 5.1. (line 388.) "We therefore come to a conclusion that part of the impact cloud often survives the passage through the photosheath to influence the antennas." It would be beneficial to provide more context or specific details to improve the text.

AC: Agreed, the sentence was slightly revised for more claritiy and a reference to Appendix G was added, since the appendix does the discussion.
* * *
RC: The conclusion of this paper could be greatly enriched by a more detailed discussion of the potential benefits and insights expected from the secondary pick. Even if these are currently based on assumptions or relate to an ongoing investigation, even a short presentation of this aspect would enhance the overall usefulness of the conclusion. Such a discussion would provide readers with a clearer perspective on the implications and possible future directions arising from the results of the secondary pick study.

AC: It is difficult to give more insight and expectations, as the nature of the scondary peaks is somewhat speculative. However, we can already give recommend to distinguish between the primary and the secondary peak, since it gives better estimates for the primary peak amplitude. The secondary peak's properties are subject to ongoing investigation and the section was expanded to mention what the secondary peak analysis may be useful for in the future.